# Defects in translation-dependent quality control pathways lead to convergent molecular and neurodevelopmental pathology

**Markus Terrey[1,2], Scott I Adamson[3,4], Jeffrey H Chuang[3,4], Susan L Ackerman[1]***

[1]Howard Hughes Medical Institute, Department of Cellular and Molecular Medicine, Section of Neurobiology, Division of Biological Sciences, University of California San Diego, La Jolla, United States; [2]Graduate School of Biomedical Sciences and Engineering, University of Maine, Orono, United States; [3]The Jackson Laboratory for Genomic Medicine, Farmington, United States; [4]Department of Genetics and Genome Sciences, Institute for Systems Genomics, UConn Health, Farmington, United States

**Abstract** Translation-dependent quality control pathways such as no-go decay (NGD), non-stop decay (NSD), and nonsense-mediated decay (NMD) govern protein synthesis and proteostasis by resolving non-translating ribosomes and preventing the production of potentially toxic peptides derived from faulty and aberrant mRNAs. However, how translation is altered and the in vivo defects that arise in the absence of these pathways are poorly understood. Here, we show that the NGD/NSD factors *Pelo* and *Hbs1l* are critical in mice for cerebellar neurogenesis but expendable for survival of these neurons after development. Analysis of mutant mouse embryonic fibroblasts revealed translational pauses, alteration of signaling pathways, and translational reprogramming. Similar effects on signaling pathways, including mTOR activation, the translatome and mouse cerebellar development were observed upon deletion of the NMD factor *Upf2*. Our data reveal that these quality control pathways that function to mitigate errors at distinct steps in translation can evoke similar cellular responses.

*For correspondence:
sackerman@ucsd.edu

Competing interests: The authors declare that no competing interests exist.

## Introduction

Regulation of gene expression is essential for cell growth and development. Although epigenetic mechanisms and transcriptional regulation are initial steps in gene expression, translation and its regulation have emerged as major hubs to control the production of functional proteins. In fact, translation is intricately coordinated with the degradation of faulty mRNAs and their resulting peptide products to balance gene expression and proteostasis (*Collart and Weiss, 2020*). Numerous studies have revealed that mRNA levels show limited correlation with protein levels, particularly when cells undergo dynamic transitions (*Abreua R de et al., 2009*; *Kristensen et al., 2013*; *Liu et al., 2016*; *Maier et al., 2009*). Indeed, post-transcriptional mechanisms, such as translation, play vital roles in rapidly altering gene expression and the signaling pathways that mediate cell identity and cell fate changes necessary for mammalian development (*Blair et al., 2017*; *Blanco et al., 2016*; *Fujii et al., 2017*; *Gabut et al., 2020*; *Kong and Lasko, 2012*; *Rodrigues et al., 2020*; *Signer et al., 2014*; *Tahmasebi et al., 2019*).

The process of translation is highly organized, and ribosomes need to accurately perform the steps of translation initiation, elongation and termination. However, multiple factors can perturb translation including secondary structures of mRNAs, amino acid limitations, tRNA deficiencies, rare

codons, interactions of nascent peptides with the ribosome, chemically damaged mRNAs, and cleaved or aberrant mRNAs (*Brandman and Hegde, 2016*; *Brule and Grayhack, 2017*; *Buhr et al., 2016*; *Drummond and Wilke, 2008*; *Hu et al., 2009*; *Simms et al., 2014*; *Spencer et al., 2012*; *Thommen et al., 2017*; *Wolf and Grayhack, 2015*; *Yu et al., 2015*). In turn, these defects may trigger translation-dependent quality control pathways including non-stop decay (NSD), no-go decay (NGD), or nonsense-mediated mRNA decay (NMD) to release ribosomes, eliminate potentially toxic or faulty peptide products, and co-translationally decay problematic or aberrant mRNAs (*Collart and Weiss, 2020*).

NSD rescues stalled ribosomes at the ends of truncated mRNAs that lack a stop codon and those in the 3'UTR of mRNAs that were not recycled at canonical stop codons (*D'Orazio et al., 2019*; *Guydosh and Green, 2014*; *Mills et al., 2017*; *Young et al., 2015*). Resolution of these stalled elongation complexes is mediated by Dom34 (yeast; PELO in mammals), and this activity is promoted in the presence of its binding partner Hbs1 (HBS1L in mammals) (*Ikeuchi et al., 2016*; *Pisareva et al., 2011*; *Shoemaker et al., 2010*; *Shoemaker and Green, 2011*; *Tsuboi et al., 2012*). In contrast, NGD resolves stalled ribosomes that are due to secondary mRNA structures, amino acid starvation or tRNA deficiency (*Ikeuchi et al., 2019*; *Inada, 2020*; *Ishimura et al., 2014*; *Simms et al., 2017b*). Defects in the resolution of stalled elongation complexes may trigger endonucleolytic cleavage of mRNAs resulting in 5'RNA intermediates that lack a stop codon and in turn, may be NSD substrates (*D'Orazio et al., 2019*; *Doma and Parker, 2006*; *Glover et al., 2020*; *Guydosh and Green, 2017*; *Inada, 2020*). Unlike NSD or NGD that utilize specialized termination factors (e.g. Dom34:Hbs1, GTPBP1, or GTPBP2 *Ishimura et al., 2014*; *Terrey et al., 2020*), NMD relies on the canonical termination factors eRF1 and eRF3 (*Dever and Green, 2012*; *Simms et al., 2017a*). UPF proteins (UPF1, UPF2, and UPF3) are critical in eliminating mRNAs that contain premature stop codons or retained introns, thus preventing generation of their potentially faulty protein products (*Karousis and Mühlemann, 2019*; *Lykke-Andersen and Bennett, 2014*; *Raimondeau et al., 2018*).

Defects in translation and translation-dependent quality control pathways impair cellular homeostasis and have been linked to proteotoxicity, changes in synaptic function, and neurodegeneration (*Choe et al., 2016*; *Chu et al., 2009*; *Huang et al., 2018*; *Ishimura et al., 2014*; *Johnson et al., 2019*; *Kapur et al., 2020*; *Martin et al., 2020*; *Notaras et al., 2020*; *Terrey et al., 2020*; *Yonashiro et al., 2016*). Here, we demonstrate that the ribosome rescue factors *Pelo* and *Hbs1l*, which are implicated in NSD and NGD, are critical for embryonic and brain development, but dispensable for neuronal survival in the adult brain. Our analysis of *Pelo*- and *Hbs1l*-deficient fibroblasts reveals translational reprogramming of multiple pathways. Inhibition of NMD via deletion of *Upf2* resulted in strikingly similar effects on the translatome, signaling pathways, and neurogenesis. Our data reveal that defects in translation-dependent quality control pathways, which mitigate errors in translation to prevent the production of defective peptide products from aberrant mRNAs, can trigger similar cellular responses and neurodevelopmental abnormalities.

## Results

### *Hbs1l* is required for embryogenesis

Multiple neurological abnormalities, including defects in motor control, were recently described in a patient with biallelic mutations in *Hbs1l* (*O'Connell et al., 2019*). Alternative splicing of *Hbs1l* produces transcripts that encode two distinct proteins (*Figure 1A*). Levels of full length *Hbs1l* (*Hbs1l*-V1 and Hbs1 in human and yeast, respectively) were dramatically decreased in *Hbs1l* patient fibroblasts (*O'Connell et al., 2019*). The levels of the shorter isoform II (*Hbs1l*-V3 in human), which is encoded by the first 4 exons of full-length *Hbs1l* and a unique last exon ('exon 5a') located between exon 4 and exon 5 of the *Hbs1l* locus, were relatively unaffected in the *Hbs1l* patient fibroblasts (*O'Connell et al., 2019*). In contrast to the translation-dependent quality control function of *Hbs1l*, previous studies suggest that isoform II of *Hbs1l* is likely an ortholog of the *Saccharomyces cerevisiae* protein SKI7 (*Brunkard and Baker, 2018*; *Kalisiak et al., 2017*; *Marshall et al., 2018*), which is involved in global mRNA turnover (*Kalisiak et al., 2017*). Although an additional splice variant (*Hbs1l*-V2 in human) which lacks the third coding exon is annotated in the human transcriptome (*Mills et al., 2016*; *O'Connell et al., 2019*), this splice variant is not annotated in mice, and we were unable to detect it by RT-PCR.

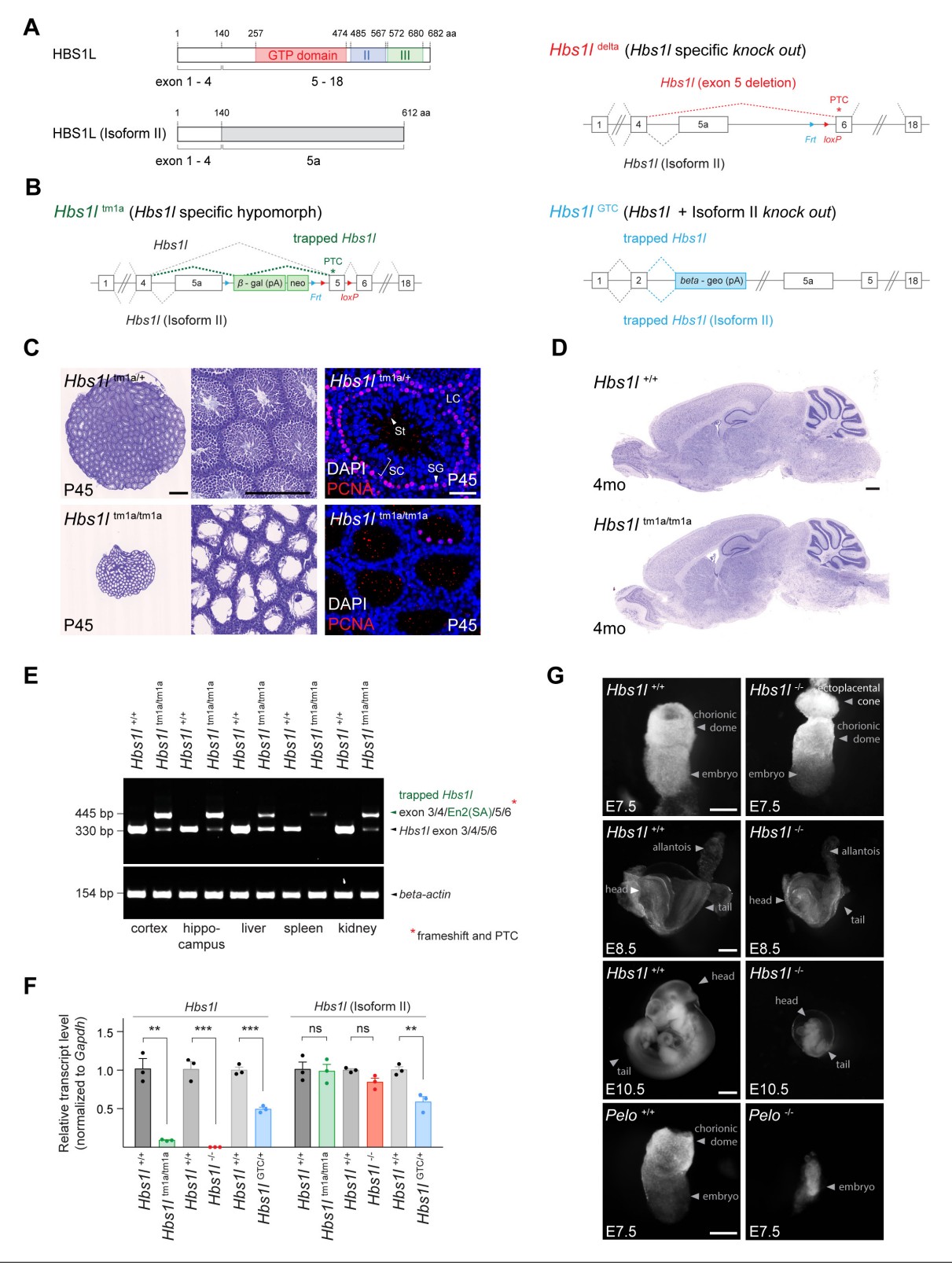

**Figure 1.** *Hbs1l* is required for embryogenesis. (**A**) Domain structure of HBS1L and isoform II and the exons encoding the two splice variants. (**B**) Design of *Hbs1l* loss-of-function alleles that target *Hbs1l* and isoform II. *Hbs1l*tm1a allele, *Hbs1l* specific gene trap (*hypomorph*); *Hbs1l*delta allele, *Hbs1l* specific deletion of exon 5 (*Hbs1l⁻*, *knock out*); and *Hbs1l*GTC allele, *Hbs1l* gene trap to target *Hbs1l* and isoform II (*knock out*). (**C**) Cresyl violet-stained cross-sections of testis from P45 control (*Hbs1l*tm1a/+) and *Hbs1l*tm1a/tm1a mice. Immunofluorescence was performed with antibodies to proliferating cell

*Figure 1 continued on next page*

*Figure 1 continued*

nuclear antigen (PCNA, red) and sections were counterstained with DAPI. (D) Cresyl violet-stained sagittal brain section from 4-month-old control ($Hbs1l^{+/+}$) and $Hbs1l^{tm1a/tm1a}$ mice. (E) Splicing analysis of correctly spliced $Hbs1l$ and trapped $Hbs1l$ transcripts in various tissues from 4-month-old control ($Hbs1l^{+/+}$) and $Hbs1l^{tm1a/tm1a}$ mice. $\beta$-actin was used as an input control. (F) Quantitative RT-PCR analysis of $Hbs1l$ and isoform II using cDNA from E8.5 embryos. Data were normalized to *Gapdh* and the fold change in gene expression is relative to that of controls ($Hbs1l^{+/+}$) from each cross. Data represent mean + SEM. (G) Bright field images of control ($Hbs1l^{+/+}$ and $Pelo^{+/+}$), $Hbs1l^{-/-}$ and $Pelo^{-/-}$ embryos at E7.5, E8.5, and E10.5. Scale bars: 500 µm and 200 µm (higher magnification), 50 µm (immunofluorescence image) (C); 500 µm (D); 100 µm (E7.5), 200 µm (E8.5), and 2 mm (E10.5) (G). PTC, premature termination codon; Frt, flippase-mediated recombination site; loxP, *Cre* recombinase-mediated recombination site; En2(SA), splice acceptor of mouse *Engrailed-2* exon 2; SC, spermatocytes; SG, spermatogonia; St, spermatids; LC, Leydig cells. t-tests were corrected for multiple comparisons using Holm-Sidak method (F). ns, not significant; **$p \leq 0.01$; ***$p \leq 0.001$.

The online version of this article includes the following source data for figure 1:

**Source data 1.** *Hbs1l* is required for embryogenesis.

To study the neurological function of *Hbs1l* in mice, we first examined an *Hbs1l* allele ($Hbs1l^{tm1a}$) with a gene trap cassette inserted between 'exon 5a' and exon 5 (*Figure 1B*). As previously reported, homozygous $Hbs1l^{tm1a/tm1a}$ mice were viable, but male mice were infertile (*O'Connell et al., 2019*). Histological analysis of the $Hbs1l^{tm1a/tm1a}$ testis at postnatal day P45 revealed a dramatic loss of mitotically active (PCNA$^+$) spermatogonia, as well as spermatocytes and spermatids that normally differentiate from these cells (*Figure 1C*). However, no overt defects were observed in the $Hbs1l^{tm1a/tm1a}$ brain (*Figure 1D*).

Residual levels of *Hbs1l* were still present in various tissues from $Hbs1l^{tm1a/tm1a}$ mice (*O'Connell et al., 2019*). In agreement, *Hbs1l* transcripts spliced into the gene trap cassette in all tested tissues; however, correctly spliced *Hbs1l* transcripts were still detected in several tissues (*Figure 1E*). Thus, to completely eliminate expression of *Hbs1l*, we ubiquitously deleted exon five in $Hbs1l^{tm1a}$ mice to generate $Hbs1l^{-/-}$ mice (*Figure 1B*). In contrast to embryonic day (E) 8.5 $Hbs1l^{tm1a/tm1a}$ embryos, which still expressed 9% of the wild-type levels of *Hbs1l* mRNA, expression of *Hbs1l* was not detected in E8.5 $Hbs1l^{-/-}$ embryos (*Figure 1F*). Expression of *Hbsl1* isoform II was not significantly changed in homozygous embryos of either allele, as predicted (*Figure 1F*). In contrast to hypomorphic $Hbs1l^{tm1a/tm1a}$ mice, $Hbs1l^{-/-}$ embryos failed to develop after E8.5 and could not be recovered at E11.5 from heterozygous matings (*Figure 1G*, *Supplementary file 1*). These results demonstrate that *Hbs1l* is necessary for embryonic development; however, embryos lacking *Hbs1l* develop longer than embryos deficient for *Pelo*, the binding partner of HBS1L, which die by E7.5 (*Adham et al., 2003*; *Figure 1G*).

To determine if isoform II of *Hbs1l* is also necessary for embryonic viability, we utilized an additional allele ($Hbs1l^{GTC}$) with a gene trap cassette located in intron 2 (*Figure 1B*). Expression of *Hbs1l* and isoform II transcripts in E8.5 heterozygous $Hbs1l^{GTC/+}$ embryos was reduced by 51% and 41%, respectively (*Figure 1F*). Homozygous $Hbs1l^{GTC/GTC}$ embryos were not recovered at E6.5 from heterozygous matings (*Supplementary file 1*) demonstrating that they died even before $Pelo^{-/-}$ or $Hbs1l^{-/-}$ embryos. The early embryonic lethality of $Hbs1l^{GTC/GTC}$ embryos suggests that the *Hbs1l* isoforms are likely functionally distinct, and that their loss causes additive or synergistic defects during embryogenesis.

## *Hbs1l* is required for cerebellar development

Consistent with transcriptome data from a brain RNA sequencing database (*Zhang et al., 2014*), we observed expression of *Hbs1l* in multiple cell types of the brain (*Figure 2A*). Expression of HBS1L and its binding partner PELO was observed throughout and after cerebellar development (*Figure 2A,B and C*). However, levels of HBS1L and PELO decreased in the postnatal (P)14 cerebellum after the completion of development, a similar decrease was observed in the whole brain (*Figure 2B and C*, *Figure 2—figure supplement 1A*). To begin to investigate the role of *Hbs1l* in the brain, we deleted this gene in the developing cerebellum and midbrain by crossing the floxed allele to $En1^{Cre}$ mice (*Kimmel et al., 2000*). Differences in cerebellar size between mutant and control embryos were already apparent by E13.5 (*Figure 2D*). Although the trilaminar structure of the cerebellum appeared normal in $En1^{Cre}$; *Hbs1l* cKO mice at postnatal day P21, cerebellar foliation was delayed in P0 mutant mice, and secondary fissures failed to form compared to control mice (*Figure 2D*). Consistent with previous studies no cerebellar abnormalities were observed in $En1^{Cre}$

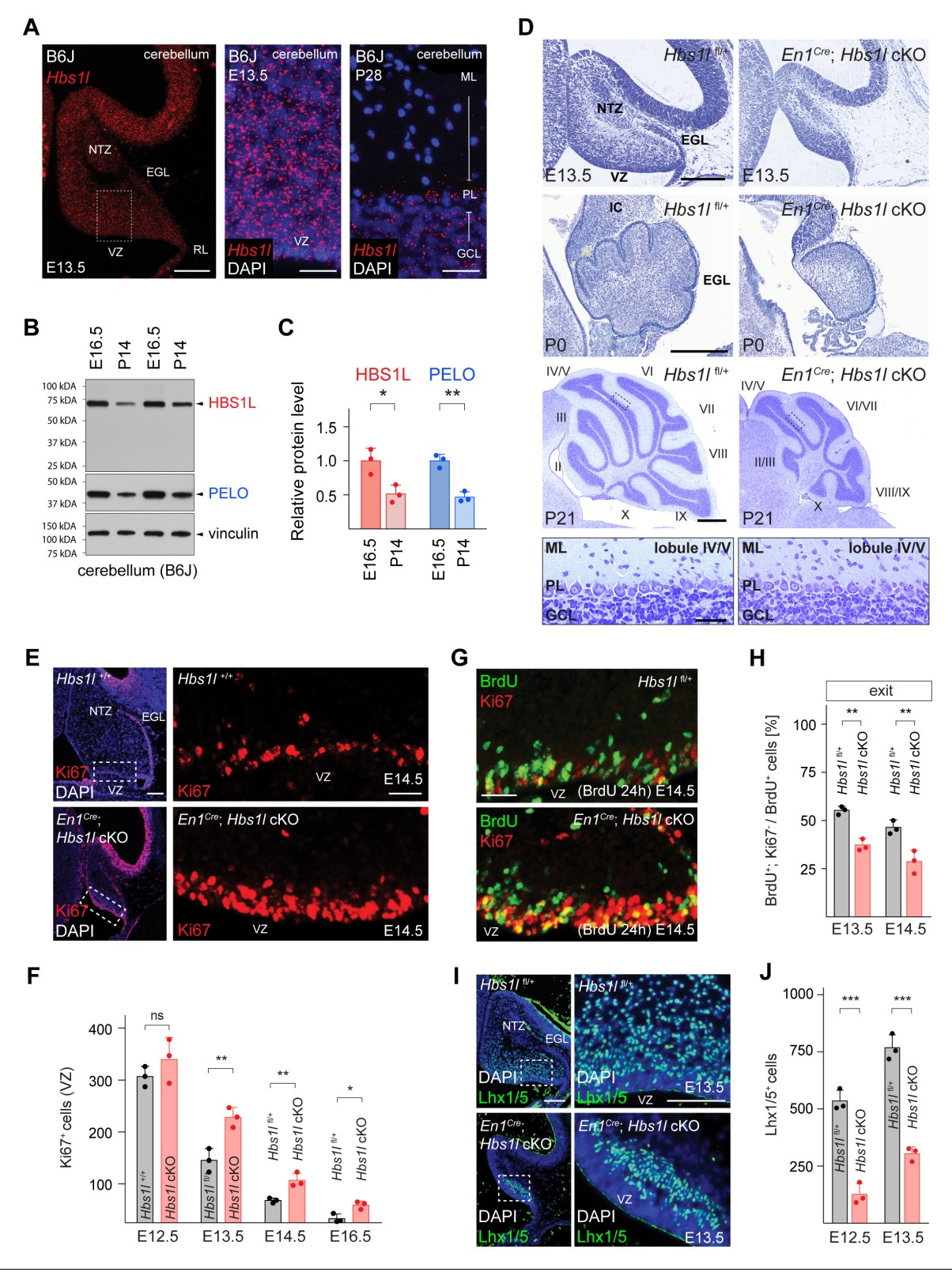

**Figure 2.** *Hbs1l* is required for cerebellar development. (**A**) In situ hybridization of *Hbs1l* mRNA (red) on cerebellar sections from E13.5 and P28 control (B6J) mice. Sections were counterstained with DAPI. (**B**) Western blot analysis using cerebellar lysates from B6J mice. Vinculin was used as a loading control. (**C**) The relative protein levels of HBS1L and PELO were normalized to levels of vinculin and protein levels are relative to those of E16.5 B6J cerebella. (**D**) Parasagittal (E13.5) and sagittal (P0 and P21) cerebellar sections from control (*Hbs1l*^fl/+^) and *En1^Cre^; Hbs1l* cKO mice stained with cresyl
*Figure 2 continued on next page*

*Figure 2 continued*

violet. Higher magnification images of lobules IV/V at P21 are shown below each genotype. Cerebellar lobules are indicated by Roman numerals. (**E**) Immunofluorescence using antibodies to Ki67 (red) on cerebellar section from E14.5 control (*Hbs1l*$^{+/+}$) and *En1*$^{Cre}$; *Hbs1l* cKO embryos. Sections were counterstained with DAPI and higher magnification images of boxed area are shown. (**F**) Number of cerebellar VZ-progenitors (Ki67$^+$ cells) from control (*Hbs1l*$^{+/+}$ or *Hbs1l*$^{fl/+}$) and *En1*$^{Cre}$; *Hbs1l* cKO embryos. Data represent mean + SD. (**G**) Immunofluorescence using antibodies to BrdU (green) and Ki67 (red) on cerebellar sections from E14.5 control (*Hbs1l*$^{fl/+}$) and *En1*$^{Cre}$; *Hbs1l* cKO embryos to determine the fraction of cells that exited the cell cycle. Embryos were injected with BrdU 24 hr prior to harvest. (**H**) Percentage of cerebellar VZ-progenitors that exited the cell cycle (BrdU$^+$, Ki67$^-$ cells). Data represent mean + SD. (**I**) Immunofluorescence using antibodies to Lhx1/5 on cerebellar sections from E13.5 control (*Hbs1l*$^{+/+}$) and *En1*$^{Cre}$; *Hbs1l* cKO embryos. Sections were counterstained with DAPI and higher magnification images of boxed areas are shown. (**J**) Number of cerebellar Purkinje cell precursors (Lhx1/5$^+$ cells). Data represent mean + SD. Scale bars: 100 µm and 20 µm (higher magnifications) (**A**); 200 µm (E13.5 and P0), 500 µm and 50 µm (higher magnification) (P21) (**D**); 100 µm and 20 µm (higher magnification) (**E**); 20 µm (**G**); 100 µm and 50 µm (higher magnification) (**I**). VZ, ventricular zone; NTZ, nuclear transitory zone; RL, rhombic lip; EGL, external granule cell layer; ML, molecular cell layer; PL, Purkinje cell layer; GCL, granule cell layer; IC, inferior colliculus. t-tests were corrected for multiple comparisons using Holm-Sidak method (**C, F, H, J**). ns, not significant; *p≤0.05; **p≤0.01; ***p≤0.001.

The online version of this article includes the following source data and figure supplement(s) for figure 2:

**Source data 1.** *Hbs1l* is required for cerebellar development.

**Figure supplement 1.** *Hbs1l* is required for the development of multiple cerebellar linages.

**Figure supplement 1—source data 1.** *Hbs1l* is required for the development of multiple cerebellar linages.

**Figure supplement 2.** *Hbs1l*$^{-/-}$-mediated cerebellar defects are independent of the B6J-associated mutation in *n-Tr20*.

**Figure supplement 2—source data 1.** *Hbs1l*$^{-/-}$-mediated cerebellar defects are independent of the B6J-associated mutation in *n-Tr20*.

mice (*Dong and Kwan, 2020*; *Guo et al., 2010*; *Li et al., 2002*; *Sgaier et al., 2007*; *Tripathi et al., 2008*).

Lineage-restricted GABAergic precursors are generated beginning at ~E10.5 from progenitors in the ventricular zone (VZ) of the developing cerebellum (*Ju et al., 2016*; *Leto et al., 2012*). The pool of progenitors declines as progenitors either exit the cell cycle to generate precursors or retract from the VZ to form a secondary germinal zone in the prospective white matter to transition from neurogenesis to gliogenesis (*Leto et al., 2012*; *Vong et al., 2015*; *Wizeman et al., 2019*). Immunofluorescence with antibodies to the cell-cycle-associated protein Ki67 demonstrated a loss of progenitors between E12.5 to E16.5 in both the control and *En1*$^{Cre}$; *Hbs1l* cKO cerebella. However, the number of progenitors remained higher in *En1*$^{Cre}$; *Hbs1l* cKO compared to control cerebella (*Figure 2E and F*), suggesting that *Hbs1l*-deficient progenitors may aberrantly proliferate. In agreement, we observed a higher fraction of VZ-progenitors in S-phase in the E12.5 and E13.5 mutant cerebella by pulse labeling with BrdU for 30 min and performing co-immunofluorescence with BrdU and Ki67 antibodies (*Figure 2—figure supplement 1B and C*). Immunofluorescence with antibodies to the M-Phase marker phospho-histone 3 (pH3) and the general cell cycle marker PCNA demonstrated an increase of VZ-progenitors in M-phase in the mutant cerebellum (*Figure 2—figure supplement 1D*).

To test if *Hbs1l*-deficient progenitors are able to exit the cell cycle, which is necessary to generate lineage-restricted precursors, we labeled control and mutant embryos at E12.5 or E13.5 with BrdU and then determined the fraction of BrdU$^+$ cells that had left the cell cycle (do not express Ki67) 24 hr after labeling. Fewer VZ-progenitors exited the cell cycle in *En1*$^{Cre}$; *Hbs1l* cKO relative to control cerebella (*Figure 2G and H*). Concordantly, the number of VZ-derived Lhx1/5$^+$ Purkinje cells and Pax2$^+$ interneuron precursors was reduced to ~32% and~34% in *En1*$^{Cre}$; *Hbs1l* cKO between E12.5 to E16.5 compared to control cerebella (*Figure 2I and J*, *Figure 2—figure supplement 1E and F*), suggesting that *Hbs1l*-deficient VZ-progenitors remain proliferative at the expense of differentiation.

Cerebellar glutamatergic precursors are generated beginning at ~E10.5 from the rhombic lip (RL), a germinal zone located in the caudal region of the cerebellar primordia. Between E10.5–12.5 the RL gives rise to Tbr1$^+$ cells that migrate subpially to take up residence in the nuclear transitory zone and will develop into deep cerebellar neurons (*Fink et al., 2006*). Following Tbr1$^+$ cell production, proliferating granule cell precursors emerge from the RL and form the external granule cell layer (EGL) (*Chung et al., 2010*). As observed for VZ-derived neuronal precursors, deletion of *Hbs1l* also decreased progeny generated from the RL. Immunofluorescence with antibodies to Tbr1, revealed fewer Tbr1$^+$ cells in the E14.5 mutant cerebella compared to controls (*Figure 2—figure*

*supplement 1G*). In addition, the EGL of *En1^Cre*; *Hbs1l* cKO at E13.5 to E16.5 contained only ~45% of the granule cell precursors compared to controls (*Figure 2—figure supplement 1H and I*).

Granule cell precursors that derive from the RL continue to proliferate in the EGL to generate additional precursors that exit the cell cycle postnatally prior to migrating to the internal granule cell layer (IGL). To determine if the decrease in granule cell precursors was due to a failure of progenitors in the RL to generate sufficient numbers of granule cell precursors, or a failure of granule cell precursors to proliferate, we labeled E12.5 and E13.5 wild type and mutant embryos with BrdU and then determined the number of BrdU⁺ cells in the EGL 24 hr after labeling. Fewer BrdU⁺ cells were present in the *En1^Cre*; *Hbs1l* cKO EGL compared to that of controls indicating that the mutant rhombic lip generates fewer granule cell precursors (*Figure 2—figure supplement 1J and K*). However, cell cycle exit of these precursors did not vary between mutant and control cells (*Figure 2—figure supplement 1L*). Furthermore, no difference in proliferation (S- and M-phase) was observed when granule cell precursors were analyzed at E16.5 or P5 (*Figure 2—figure supplement 1M and N*). Together, these data suggest that while *Hbs1l* is required for the initial production of granule cell precursors, it is dispensable for the subsequent cell cycle progression of these precursors in the EGL. In agreement, genetic ablation of *Hbs1l* by Tg(Atoh1-Cre) which specifically deletes in cerebellar granule cell precursors in the EGL beginning at ~E16.5 (*Lorenz et al., 2011*; *Pan et al., 2009*; *Qiu et al., 2010*; *Wojcinski et al., 2019*), did not impair development of these cells (*Figure 2—figure supplement 2A*).

*Hbs1l* was also required for gliogenesis in the developing cerebellum. Cerebellar gliogenesis starts at ~E18 and continues during postnatal development during which time progenitors that have retracted from the VZ switch from a neurogenic to gliogenic fate and produce oligodendrocytes and astrocytes (*Götz and Huttner, 2005*; *Vong et al., 2015*). Immunofluorescence with antibodies to Olig2, a marker of oligodendroglial progenitors which give rise to oligodendrocytes and astrocytes (*Chung et al., 2013*; *Tatsumi et al., 2018*), revealed the number of these cells was reduced in P5 *En1^Cre*; *Hbs1l* cKO cerebella (*Figure 2—figure supplement 1O*). Together these data indicate that *Hbs1l* is required for the generation of multiple cell types in the developing cerebellum.

We have previously identified a mutation in the common C57BL/6J (B6J) strain that partially disrupts processing of the brain-specific arginine tRNA, *n-Tr20* (*n-Tr20^B6J/B6J*; *n-Tr20* is also known as *n-TRtct5*). This processing defect in turn reduces the pool of available tRNA^Arg_UCU, leading to ribosome pausing at the A-site at AGA codons in cerebellar mRNAs (*Ishimura et al., 2014*). *Hbs1l* cKO mice were generated with the B6J-associated mutation in *n-Tr20*. To test if the *n-Tr20* deficiency influenced the developmental defects observed in the absence of *Hbs1l*, we either restored *n-Tr20* to wild type levels or completely deleted *n-Tr20* in *Hbs1l* cKO mice. Wild-type expression of the tRNA (*n-Tr20^B6N/B6N*) did not rescue defects in *En1^Cre*; *Hbs1l* cKO cerebella, nor did complete loss of *n-Tr20* (*n-Tr20^-/-*) cause developmental defects in Tg(Atoh1-Cre); *Hbs1l* cKO cerebella (*Figure 2—figure supplement 2A*). In addition, neither loss of *Hbs1l* or *Pelo* affected cell survival of terminally differentiated granule cells in 9-month-old mice even in the presence of the *n-Tr20* deficiency (*Figure 2—figure supplement 2B,C,D and E*), indicating that *Hbs1l* and *Pelo* do not respond to AGA pausing.

## Ribosome pausing correlates with pathology in *Hbs1l*-deficient mice

To determine if the developmental defects that occur upon loss of *Hbs1l* are accompanied by alterations in translation elongation, we performed ribosome profiling on wild type and *Hbs1l^-/-* embryos at E8.5. Ribosome protected fragments (RPF) mapped primarily to the protein coding sequence of genes in both wild-type and mutant embryos (*Figure 3A*). Using the previously described methodology (*Ishimura et al., 2014*), we found a total of ~1300 sites with significant (z-score ≥10) increases in local ribosome occupancy ('ribosome pauses') in wild-type and mutant embryos (*Figure 3B*, *Supplementary file 2*). About 40% of the ribosome pauses, which mapped to 319 genes, were shared between genotypes suggesting they occurred independently of the loss of *Hbs1l*. Ten percent of ribosome pauses (mapped to 107 genes) were found only in wild-type embryos. The ribosome density derived from the total number of ribosome-protected fragments (RPF) serves as proxy for gene expression at the level of the translatome. Differential expression analysis of the translatome (DE RPF, *Supplementary file 3*) indicated translation of genes with wild type-specific pauses was decreased in mutant embryos (*Figure 3C*), which may contribute to the apparent specificity of these pauses to wild-type embryos. Strikingly, 50% of ribosome pauses (mapped to 459 genes) were

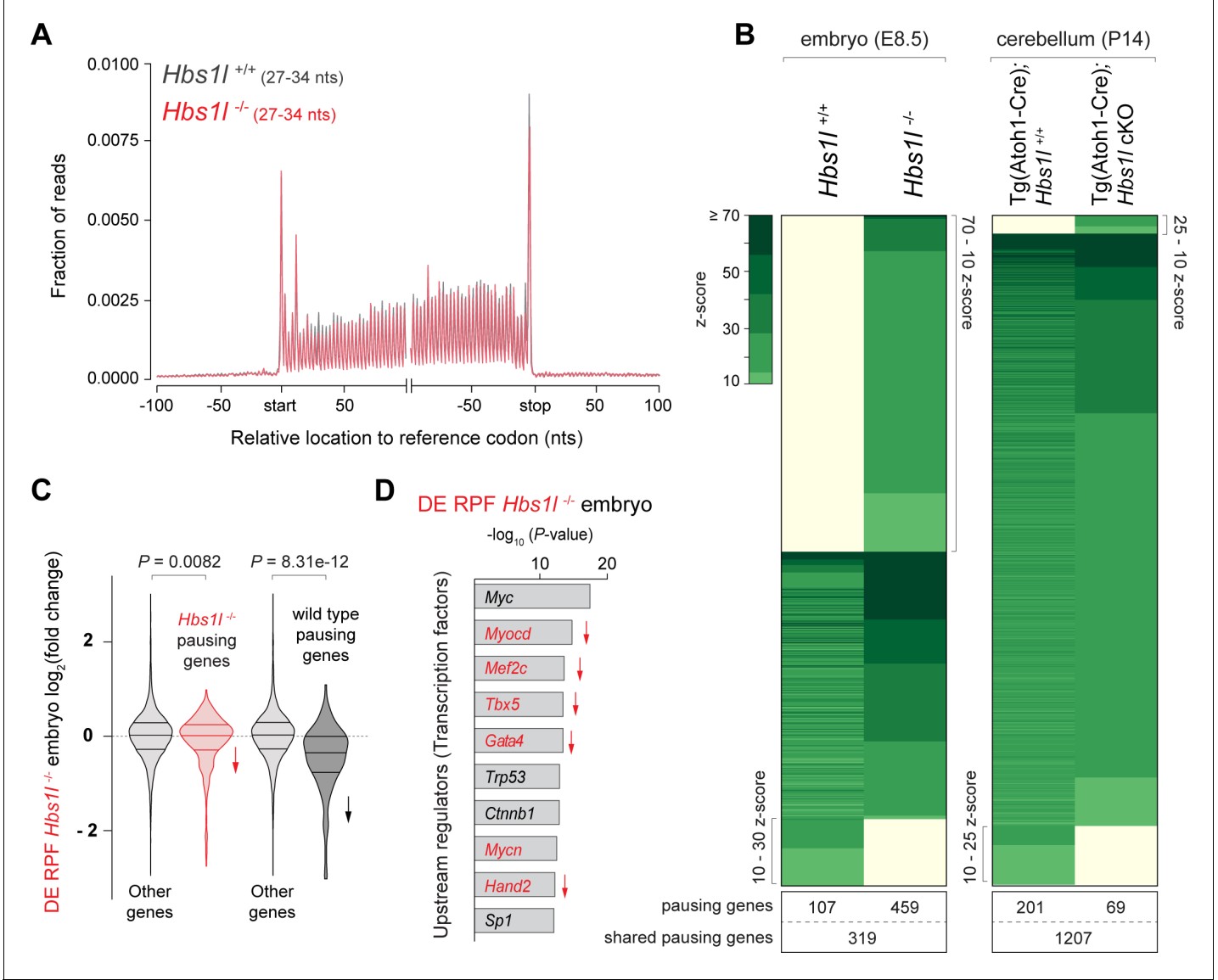

**Figure 3.** Ribosome pausing correlates with pathology in *Hbs1l*-deficient mice. (A) Metagene profiles of RPFs from E8.5 control (*Hbs1l*⁺/⁺, gray traces) and *Hbs1l*⁻/⁻ (red traces) embryos. (B) Analysis of significantly increased local ribosome occupancy (z-score ≥10, pause site detected in all three replicates) from E8.5 wild type (*Hbs1l*⁺/⁺) and *Hbs1l*⁻/⁻ embryos (left) or P14 control (Tg(Atoh1-Cre); *Hbs1l*⁺/⁺) and Tg(Atoh1-Cre); *Hbs1l* cKO cerebella (right). The number of genes that pause sites map to is shown below for each genotype. (C) All translated genes (DE RPF *Hbs1l*⁻/⁻) from E8.5 *Hbs1l*⁻/⁻ embryos were compared to the translation of genes which contained pauses specific to either *Hbs1l*⁻/⁻ or wild-type embryos. Downward direction of arrows indicates significant reduction in translation of pausing genes in E8.5 *Hbs1l*⁻/⁻ embryos relative to wild-type embryos. (D) Identification of upstream regulators using Ingenuity Pathway Analysis (IPA) of differentially translated genes of E8.5 *Hbs1l*⁻/⁻ embryos (DE RPF *Hbs1l*⁻/⁻). Transcription factors that are involved in heart development are shown in red. Downward direction of arrows indicates predicated activity (downregulation) of transcription factors. RPFs, ribosome-protected fragments; nts, nucleotides. Wilcoxon rank-sum test was used to determine statistical significance (C). The online version of this article includes the following figure supplement(s) for figure 3:

**Figure supplement 1.** *Hbs1l* deficiency in embryos alters translation of pathways associated with heart function.

uniquely observed in *Hbs1l*⁻/⁻ embryos and like other pauses, occurred primarily in protein coding sequences (*Supplementary file 2*). Similar to the metagene analysis (*Figure 3A*), we didn't observe genotype-dependent differences in the ribosome occupancy in different gene regions (i.e. untranslated 5' or 3' region). Only 9% of genes associated with *Hbs1l*⁻/⁻-specific ribosome pauses were differentially translated in *Hbs1l*⁻/⁻ embryos (DE RPF *Hbs1l*⁻/⁻) and translation of 26 genes and 15 genes was decreased and increased, respectively (*Figure 3—figure supplement 1A*).

Loss of *Hbs1l* did not affect survival of granule cells (*Figure 2—figure supplement 2B and C*), which constitute the vast majority of the cellular content of the cerebellum. Thus, to determine if elongation defects correlate with pathogenesis in *Hbs1l* mutant tissues, we also performed ribosome profiling on the cerebellum of P14 control and Tg(Atoh1-Cre); *Hbs1l* cKO mice (*Supplementary file 2*). We observed more ribosome pausing sites (~2700) in the cerebellum than in embryos, likely due to the higher amount of input RNA and sequencing depth. Consistent with the lack of a genetic interaction between the *Hbs1l* and the *n-Tr20* mutation, no significant increase in ribosome occupancy on A-site AGA codons was observed in *Hbs1l*$^{-/-}$ cerebella (*Supplementary file 4*). Unlike the high percentage of pauses that were unique to *Hbs1l*$^{-/-}$ embryos, only 2% of ribosome pauses (mapped to 69 genes) were unique to the mutant cerebellum (*Figure 3B*). In addition, the z-scores ('pause scores') for *Hbs1l*-specific pauses were significantly higher in mutant embryos than the mutant cerebellum (*Figure 3—figure supplement 1B*).

Mutant embryos also exhibited larger changes in the translatome (DE RPF) than the mutant cerebellum (*Figure 3—figure supplement 1C*), indicating that defects in translation also correlate with pathology. Kyoto Encyclopedia of Genes and Genomes (KEGG) and Ingenuity Pathway Analysis (IPA) of differentially translated genes (DE RPF *Hbs1l*$^{-/-}$, adj. p≤0.05) in *Hbs1l*$^{-/-}$ embryos revealed that downregulated genes were significantly enriched for heart/cardiac muscle contraction and calcium signaling (*Figure 3—figure supplement 1D and E*). In agreement, upstream regulator analysis predicted downregulation of multiple transcription factors required for heart development (*Cui et al., 2018*; *Molkentin et al., 1997*; *Muñoz-Martín et al., 2019*; *Steimle and Moskowitz, 2017*; *Figure 3D*). Together these data suggest that HBS1L deficiency may cause defects in the embryonic heart, one of the first organs to begin developing in the mouse embryo.

## Loss of *Pelo/Hbs1l* alters translation regulation and reprograms the translatome

Our embryonic data suggested that defects in translation modulate the translatome. Because *Pelo*$^{-/-}$ embryos could not be profiled due to their early embryonic lethality, we conditionally deleted *Pelo* or *Hbs1l* in primary mouse embryonic fibroblasts (MEFs) using a tamoxifen-inducible *Cre* transgene (TgCAG-Cre$^{ER}$) to compare changes in translation upon loss of *Pelo* and *Hbs1l* (*Figure 4A*). Consistent with previous studies (*Juszkiewicz et al., 2020*; *O'Connell et al., 2019*), deletion of *Hbs1l* was accompanied by decreased levels of PELO protein, but not its mRNA (*Figure 4B*, *Figure 4—figure supplement 1A,B and C*). In addition, we found deletion of *Pelo* also led to the loss of HBS1L without altering *Hbs1l* mRNA levels, suggesting degradation of PELO or HBS1L protein occurs in the absence of either interacting partner (*Figure 4B*, *Figure 4—figure supplement 1A,B and C*).

To analyze defects in translation elongation, we performed ribosome profiling of tamoxifen-treated control (TgCAG-Cre$^{ER}$), *Pelo*$^{-/-}$ (TgCAG-Cre$^{ER}$; *Pelo* cKO), and *Hbs1l*$^{-/-}$ (TgCAG-Cre$^{ER}$; *Hbs1l* cKO) cells. Analyzing the ribosome occupancy in *Pelo*$^{-/-}$ and control cells revealed ~10,000 sites with significant (z-score ≥10) increases in local ribosome occupancy which mapped to 4693 genes (*Figure 4C*, *Supplementary file 2*). One percent of these ribosome pauses were only observed in control cells ('control-pauses'), 57% were shared between control and *Pelo*$^{-/-}$ cells, and 42% were specific to *Pelo*$^{-/-}$ cells ('*Pelo*$^{-/-}$-pauses'). In contrast, analysis of *Hbs1l*$^{-/-}$ and control cells revealed fewer ribosome pauses (~4200 - mapping to 1807 genes) and only 5% were specific to *Hbs1l*$^{-/-}$ cells ('*Hbs1l*$^{-/-}$-pauses') (*Figure 4C*). In addition to the fewer ribosome pause sites, the z-scores ('pause scores') for *Hbs1l*$^{-/-}$-pauses were also significantly lower than those of *Pelo*$^{-/-}$-pauses (*Figure 4D*). Approximately 80% of the *Hbs1l*$^{-/-}$-pausing genes also had pauses in *Pelo*$^{-/-}$ cells; however, only 30% of the ribosome pauses occurred at the same pause site (*Figure 4E*, *Supplementary file 2*). Together, these data suggest that the loss of *Pelo* or *Hbs1l* leads to translation elongation defects, but particularly severe defects are observed in the absence of *Pelo*.

The Dom34:Hbs1 complex has been implicated in multiple translation-dependent quality control pathways including non-stop decay (NSD) (*Collart and Weiss, 2020*; *Simms et al., 2017a*). This pathway rescues ribosomes stalled at the ends of truncated mRNAs, ribosomes in polyA-sequences on prematurely polyadenylated mRNAs that lack a termination codon, or ribosomes in the 3'UTR of mRNAs that were not recycled at canonical stop codons (*Arribere and Fire, 2018*; *D'Orazio et al., 2019*; *Guydosh and Green, 2014*; *Mills et al., 2017*; *Young et al., 2015*). Examination of *Pelo*$^{-/-}$- and *Hbs1l*$^{-/-}$-pauses revealed that ~92% of pauses mapped to the protein-coding region of transcripts (*Supplementary file 5*). The remaining local increases in ribosome occupancy were similarly

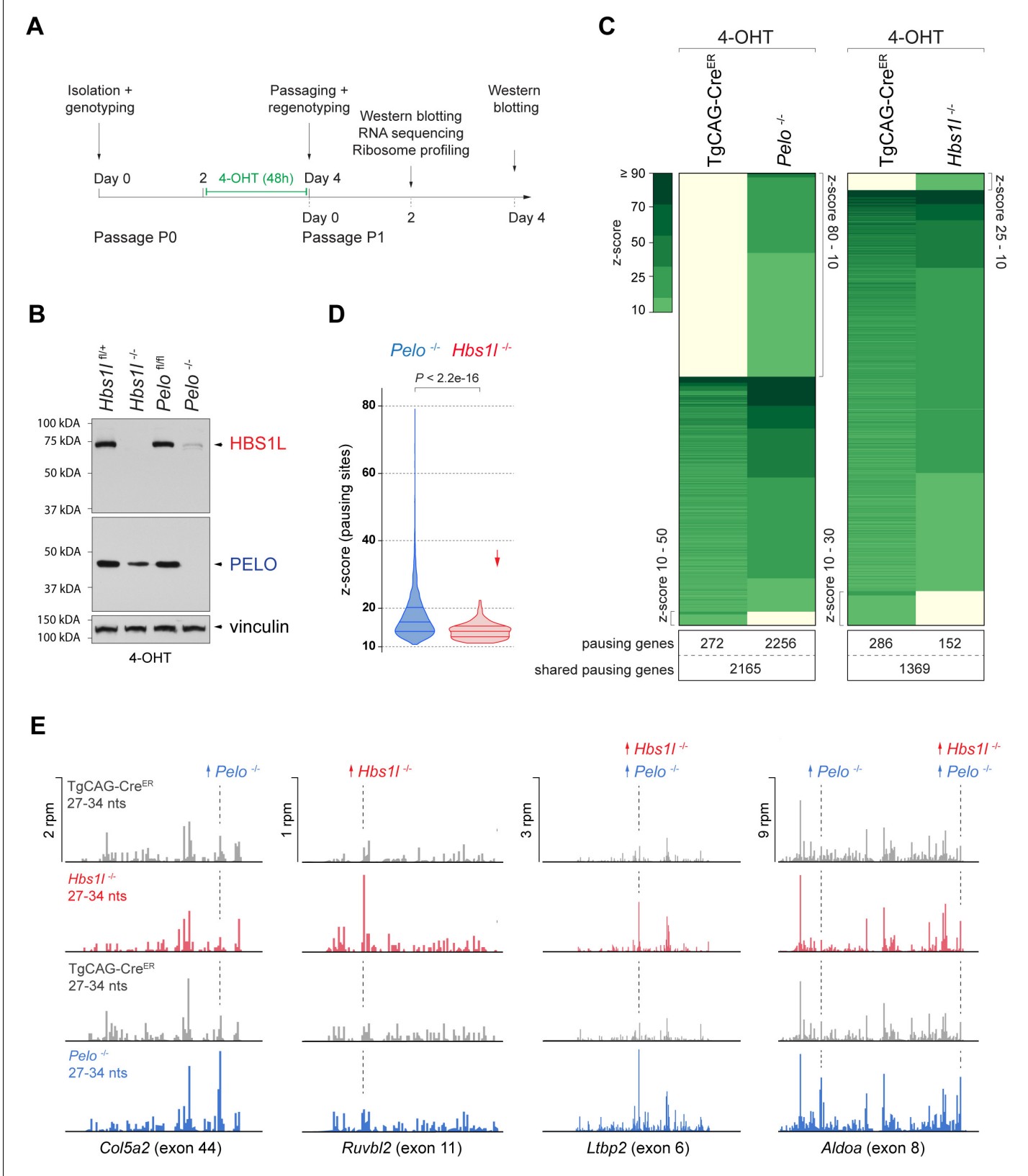

**Figure 4.** Loss of *Pelo* induces greater defects in translation elongation then loss of *Hbs1l*. (**A**) Experimental strategy for in vitro studies using tamoxifen (4-OHT) treatment of primary mouse embryonic fibroblasts (MEFs). (**B**) Western blot analysis of HBS1L and PELO using MEF lysates from tamoxifen-treated control (*Hbs1l*^fl/+^ or *Pelo*^fl/fl^), *Hbs1l*^-/-^ and *Pelo*^-/-^ cells. Vinculin was used as a loading control. (**C**) Analysis of significantly increased local

*Figure 4 continued on next page*

*Figure 4 continued*

ribosome occupancy (z-score ≥10, pause site detected in all three replicates) from tamoxifen-treated control (TgCAG-Cre<sup>ER</sup>) and *Pelo*<sup>-/-</sup> (left), or control and *Hbs1l*<sup>-/-</sup> cells (right). The number of genes that pause sites map to is shown below for each genotype. (D) Comparison of the z-scores ('pause score') for pauses observed in *Pelo*<sup>-/-</sup> (blue) and *Hbs1l*<sup>-/-</sup> (red) cells. Downward direction of the arrow indicates significant lower pause scores of *Hbs1l*<sup>-/-</sup>- compared to *Pelo*<sup>-/-</sup>-specific pauses. (E) Examples of mapped footprints (27–34 nucleotides) on genes from tamoxifen-treated control (TgCAG-Cre<sup>ER</sup>, gray) and *Pelo*<sup>-/-</sup> (blue), or control and *Hbs1l*<sup>-/-</sup> (red) cells. Upward direction of arrows indicates significant increase in local ribosome occupancy and the dashed line indicates the pause site. 4-OHT, 4-hydroxytamoxifin; nts, nucleotides; rpm, reads per million. Wilcoxon rank-sum test was used to determine statistical significance (D).

The online version of this article includes the following source data and figure supplement(s) for figure 4:

**Figure supplement 1.** Loss of *Hbs1l* and *Pelo* control protein levels of their respective binding partner.

**Figure supplement 1—source data 1.** Loss of *Hbs1l* and *Pelo* control protein levels of their respective binding partner.

split between the 5' and 3'UTRs. Although the frequency of pauses in the 3'UTR was similar for mutant-specific, control-specific and shared pauses, the frequency of control-specific pauses that mapped to the 5'UTR was increased and those in the coding region were decreased (*Figure 4—figure supplement 1D*). Thus, consistent with our observations in *Hbs1l*<sup>-/-</sup> embryos, these data suggest that neither loss of *Hbs1l* nor *Pelo* in MEFs led to enrichment of ribosomes in the 5' or 3'UTR.

We also searched for RPFs containing untemplated stretches of adenosines (A) at the 3'end indicative of ribosomes extending into the poly(A) tail of premature polyadenylated mRNAs. Ribosomes did not protect more then 15 consecutive A's (*Figure 4—figure supplement 1E*) suggesting that like in yeast, the poly(A) tract does not extend beyond the P-site of the ribosome (*Guydosh and Green, 2014*). Interestingly, we observed a significantly higher fraction of these 3'end A reads in *Pelo*<sup>-/-</sup> compared to control cells (*Figure 4—figure supplement 1F*), supporting a role for *Pelo* in rescuing ribosomes in polyA tails (*D'Orazio et al., 2019*; *Guydosh and Green, 2017*; *Guydosh and Green, 2014*). However, the fraction of those reads did not significantly increase in *Hbs1l*<sup>-/-</sup> MEFs (*Figure 4—figure supplement 1F*), *Hbs1l*<sup>-/-</sup> embryos (Student's t-test, p=0.4621) or *Hbs1l*<sup>-/-</sup> cerebella (Student's t-test, p=0.3269).

In addition to NSD, Dom34:Hbs1 is implicated in ribosome rescue during No-Go decay (NGD), a quality control pathway in which ribosome stalling is evoked due to stretches of rare codons, secondary mRNA structures, amino acid starvation, or tRNA deficiency. NGD may trigger endonucleolytic cleavage of mRNAs due to ribosome collision, leading to 5'RNA intermediates that lack a stop codon and are then targeted by Dom34:Hbs1 via NSD (*D'Orazio et al., 2019*; *Doma and Parker, 2006*; *Glover et al., 2020*). Studies in *Drosophila* and *C. elegans* revealed that RNA intermediates that converge onto NSD may also be generated through additional mechanisms including RNA interference (RNAi) and nonsense-mediated decay (NMD) (*Arribere and Fire, 2018*; *Hashimoto et al., 2017*).

To determine if loss of *Pelo* or *Hbs1l* alters the frequency of ribosome pauses that are associated with potential targets such as NMD targets, we mapped pauses to unique coding transcripts, which is generally difficult due to the short nature of RPFs. Indeed, only a fraction (~30%) of *Pelo*<sup>-/-</sup> and *Hbs1l*<sup>-/-</sup>-pauses could be assigned to unique coding transcripts (1518 and 95, respectively) (*Supplementary file 5*). Of these transcripts, ~8% were classified as NMD transcripts and ~92% were protein-coding transcripts. A similar percentage of pauses mapping to NMD transcripts (~6.5%) was observed when we analyzed transcripts that contained ribosome pauses shared between either control and *Pelo*<sup>-/-</sup> cells, control and *Hbs1l*<sup>-/-</sup> cells, or control-specific pausing transcripts (*Figure 4—figure supplement 1G*). In addition, transcriptome analysis of MEFs revealed that NMD transcripts represented ~6% of expressed transcripts (*Supplementary file 7*). These data suggest that loss of *Pelo* or *Hbs1l* did not lead to enrichment of ribosome pauses on transcripts predicted to undergo NMD.

Our observation that ribosome pausing is more dramatic in *Pelo*<sup>-/-</sup> than *Hbs1l*<sup>-/-</sup> cells correlates with the earlier lethality of *Pelo*<sup>-/-</sup> embryos. To get a broader perspective of whether the loss of *Pelo* impacts other aspects of translation more than the *Hbs1l* deficiency, we analyzed at first the translational efficiency (TE, *Supplementary file 6*) of genes by normalizing the abundance of ribosomal footprint reads to that of the RNA sequencing reads. In *Pelo*<sup>-/-</sup> cells about 35% (4884) of genes displayed significant (adj. p≤0.05) alterations in TE compared to control cells. The TE of 57% of these genes was increased and for 43% it was decreased. In contrast, the TE of only 4% of genes (314,

60% up- and 40% downregulated) was significantly changed in $Hbs1l^{-/-}$ cells and genes with differential translational efficiency showed only a moderate correlation between $Hbs1l^{-/-}$ and $Pelo^{-/-}$ cells (Pearson's correlation, $r = 0.3739$). The TE of $Pelo^{-/-}$-specific pausing genes increased, but this effect was less for $Hbs1l^{-/-}$-specific pausing genes (*Figure 5—figure supplement 1A*), suggesting that elongation defects maybe more dramatic in $Pelo^{-/-}$ cells as a result of the increase in translation of these genes.

In addition to its greater impact on the TE of genes, the loss of *Pelo* also had a stronger impact on gene translation in general as evidenced by the differential expression of ribosome-protected fragments (DE RPF, *Supplementary file 3*; *Figure 5—figure supplement 1B*). While *Hbs1l* deficiency altered translation of 26% (3445) of genes, the loss of *Pelo* affected translation of 34% (4965) of genes and led to significantly greater fold changes in gene expression in $Pelo^{-/-}$ cells (Wilcoxon test, p<2.2e-16). However, differentially translated genes between $Pelo^{-/-}$ (DE RPF $Pelo^{-/-}$) and $Hbs1l^{-/-}$ (DE RPF $Hbs1l^{-/-}$) cells were strongly correlated (*Figure 5—figure supplement 1C*), indicating that, although loss of *Hbs1l* may lead to smaller alterations in translation, most of the changes occur in the same direction.

In addition, 20% (1060) and 2% (70) of the differentially translated genes in $Pelo^{-/-}$ and $Hbs1l^{-/-}$ cells also contained ribosome pauses specifically found in $Pelo^{-/-}$ and $Hbs1l^{-/-}$ cells, respectively, and translation of these pausing genes was both increased and decreased (*Figure 5—figure supplement 1B*). However, changes in expression were under 2-fold for the majority of $Pelo^{-/-}$- and $Hbs1l^{-/-}$-pausing genes (88% of $Pelo^{-/-}$- and 93% of $Hbs1l^{-/-}$-pausing genes) and these changes were significantly lower compared to those in genes without mutant-specific pauses (*Figure 5—figure supplement 1B and D*). Thus, similar to $Hbs1l^{-/-}$ embryos, these data suggest that most of the translational changes in gene expression are not due to an increase in the ribosome occupancy but are likely a response to changes in mRNA expression and/or translation regulation.

Interestingly, we observed an opposing relationship between transcriptional expression changes (DE mRNA) and changes in translational efficiency (TE) of many genes in $Pelo^{-/-}$ and $Hbs1l^{-/-}$ cells (*Figure 5A*). KEGG pathway analysis of genes with this opposing behavior revealed enrichment of several pathways (ribosome, ribosome biogenesis, RNA transport, spliceosome, cell cycle, and lysosome) that overlapped with those of differently translated genes (DE RPF) (*Figure 5B*, *Figure 5—figure supplement 1E*), suggesting that expression of genes in these pathways is translationally regulated, perhaps to restore homeostasis between the transcriptome and translatome.

To identify signaling pathways that might control these changes in translation regulation upon loss of *Pelo/Hbs1l*, we performed IPA analysis on differentially transcribed (DE mRNA) genes and genes with altered translation efficiency (TE). EIF2 and mTOR/p70S6K signaling, both of which are known to regulate translation, were highly enriched in $Pelo^{-/-}$ cells but less enriched in $Hbs1l^{-/-}$ cells (*Figure 5C*, *Figure 5—figure supplement 1F*). Phosphorylation of eIF2α decreases translation initiation, while the activity of mTOR, in particular mTORC1 (mechanistic target of rapamycin complex 1), increases translation initiation and elongation. About 50% of the differentially regulated genes identified by IPA overlapped between the two pathways and therefore, we assessed the phosphorylation status of p-eIF2α$^{S51}$ and p-p70S6$^{T389}$, a known target of mTORC1, to determine if one or both signaling pathways were affected. Levels of p-eIF2α$^{S51}$ in $Pelo^{-/-}$ cells were unchanged from those of control cells (*Figure 5D and E*). However, levels of p-p70S6$^{T389}$ were significantly increased in $Pelo^{-/-}$ cells, indicating activation of mTOR signaling (*Figure 5D and E*). In agreement, the TE of genes that are known to be translationally regulated by mTORC1 via their 5'terminal oligopyrimidine motifs (5'TOP) (*Yamashita et al., 2008*) was significantly increased in $Pelo^{-/-}$ and $Hbs1l^{-/-}$ cells, although this increase was less pronounced in the latter (*Figure 5F*). Activation of mTORC1 may underlie some of the observed gene expression changes (DE RPF) given its role as a positive regulator for ribosome biogenesis and translation of ribosomal genes and a negative regulator of lysosomal biogenesis and autophagy (*Kim et al., 2011*; *Mayer and Grummt, 2006*; *Puertollano, 2014*; *Rabanal-Ruiz et al., 2018*; *Roczniak-Ferguson et al., 2012*), which parallels the directionality of the changes in these pathways (*Figure 5—figure supplement 1E*).

## Convergent modulation of the translatome in cells with defects in translation-dependent quality control pathways

Our findings suggest that translational reprogramming occurs upon loss of *Pelo/Hbs1l*. However, whether these changes are unique to *Pelo/Hbs1l* or reflect a more general cellular response upon

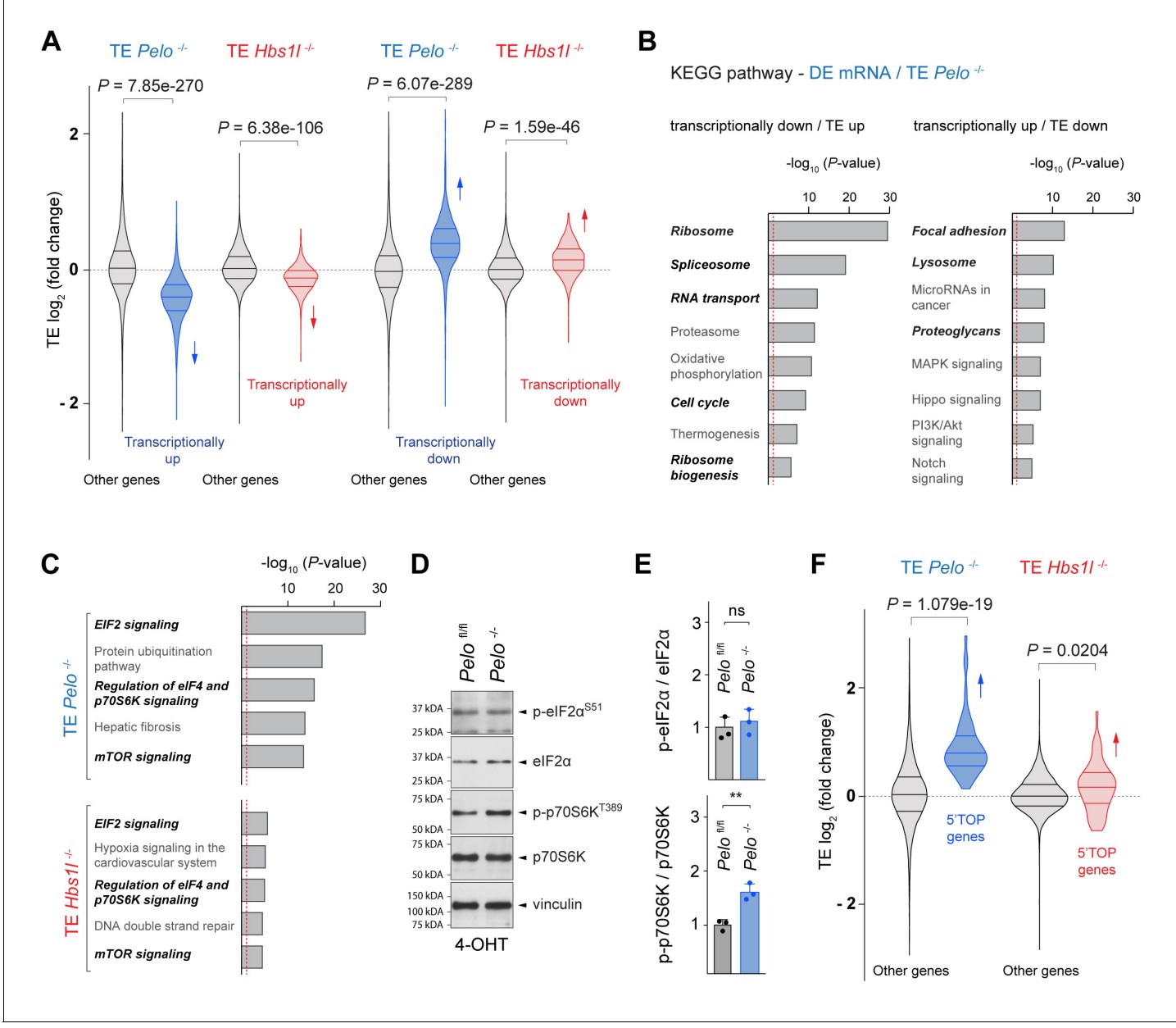

**Figure 5.** *Pelo/Hbs1l* deficiency alters translation regulation and reprograms the translatome. (**A**) Translational efficiency (TE) of genes that are transcriptionally upregulated or downregulated was compared to the remaining ('other') genes from *Pelo⁻/⁻* (blue) and *Hbs1l⁻/⁻* (red) MEFs, respectively. Downward direction of arrows indicates significant decrease in translational efficiency of transcriptionally upregulated genes in *Pelo⁻/⁻* and *Hbs1l⁻/⁻* MEFs. Upward direction of arrows indicates significant increase in translational efficiency of transcriptionally downregulated genes in *Pelo⁻/⁻* and *Hbs1l⁻/⁻* MEFs. (**B**) KEGG pathway analysis of genes from *Pelo⁻/⁻* MEFs in which the translational efficiency (TE *Pelo⁻/⁻*, adj. p≤0.05) is opposite to their transcriptional expression (DE mRNA *Pelo⁻/⁻*, *q*-value ≤0.05). Italicized pathways indicate pathways that overlapped with enriched pathways of differentially translated genes in *Pelo⁻/⁻* MEFs (DE RPF *Pelo⁻/⁻*, adj. p≤0.05) (**Figure 5—figure supplement 1E**). The red dashed line indicates the significance threshold (p=0.05). (**C**) Ingenuity pathway analysis (IPA) of genes with differential translational efficiency from *Pelo⁻/⁻* (TE *Pelo⁻/⁻*) and *Hbs1l⁻/⁻* (TE *Hbs1l⁻/⁻*) MEFs. EIF2 and mTOR/p70S6K signaling are in italics. The red dashed line indicates the significance threshold (p=0.05). (**D**) Western blot analysis of p-eIF2αˢ⁵¹ and p-p70S6Kᵀ³⁸⁹ using lysates of tamoxifen-treated control (*Pelo*ᶠˡ/ᶠˡ) and *Pelo⁻/⁻* MEFs at Day 2 (Passage P1). Vinculin was used as a loading control. (**E**) Levels of p-eIF2αˢ⁵¹ or p-p70S6Kᵀ³⁸⁹ were normalized to total level of eIF2α or p70S6K, and phosphorylation levels are relative to those of control (*Pelo*ᶠˡ/ᶠˡ). Data represent mean + SD. (**F**) Translational efficiency (TE) of genes with translational regulation by mTOR via their 5'TOP motif was compared to the remaining ('other') genes from *Pelo⁻/⁻* (blue) and *Hbs1l⁻/⁻* (red) MEFs. Upward direction of arrows indicates significant increase in translational efficiency of 5'TOP genes. 4-OHT, 4-hydroxytamoxifin; 5'TOP, 5'terminal oligopyrimidine motif. Student's t-test (**E**); Wilcoxon rank-sum test was used to determine statistical significance (**A, F**). ns, not significant; **p≤0.01.

The online version of this article includes the following source data and figure supplement(s) for figure 5:

*Figure 5 continued on next page*

*Figure 5 continued*

**Source data 1.** *Pelo/Hbs1l* deficiency alters translation regulation and reprograms the translatome.

**Figure supplement 1.** *Pelo/Hbs1l* deficiency alters translational gene expression of multiple pathways.

impairment of translation-dependent quality control pathways is unclear. To investigate this possibility, we conditionally deleted the core NMD component, *Upf2*, in MEFs (*Lelivelt and Culbertson, 1999*; *Serin et al., 2001*). In contrast to NGD and NSD that resolve ribosomes on mRNAs impeding translation elongation, nonsense-mediated decay (NMD) targets aberrant mRNAs (e.g. mRNAs containing a premature termination codon) for degradation during translation (*Karousis and Mühlemann, 2019*; *Schuller and Green, 2018*). Consistent with the function of *Upf2* in NMD, about 20% and 13% of upregulated transcripts (DE mRNA *Upf2*, q-value ≤0.05) were NMD transcripts or transcripts with retained introns, respectively (*Figure 6—figure supplement 1A*, *Supplementary file 7*). Although the accumulation of these transcripts was specific to *Upf2* $^{-/-}$ cells (*Figure 6—figure supplement 1A and B*), we observed an inverse relationship between transcriptional gene expression changes (DE mRNA *Upf2*) and changes in translational efficiency (TE *Upf2*) in *Upf2*$^{-/-}$ cells as we did in *Pelo*$^{-/-}$ and *Hbs1l*$^{-/-}$ cells (*Figure 6A*). Surprisingly, IPA analysis on differentially transcribed (DE mRNA *Upf2*$^{-/-}$) genes and genes with altered translation efficiency (TE *Upf2*$^{-/-}$) revealed enrichment for EIF2 and mTOR signaling in *Upf2*$^{-/-}$ as observed in *Pelo*$^{-/-}$ and *Hbs1l*$^{-/-}$ cells (*Figure 6B and C*). Western blot analysis revealed the level of p-eIF2α$^{S51}$ in *Upf2*$^{-/-}$ cells was unchanged from that of controls, but p-p70S6$^{T389}$ levels and translation of 5'TOP genes were significantly increased (*Figure 6C,D and E*), indicating that mTORC1 is activated in *Upf2*$^{-/-}$ cells. Although translation of ribosomal genes was increased, transcriptional levels of ribosomal genes were decreased in *Upf2*$^{-/-}$ similar to *Pelo*$^{-/-}$ cells (*Figure 6F*), further supporting that mTORC1 activation may be a general response in an attempt to restore cellular homeostasis in *Upf2*$^{-/-}$ and *Pelo*$^{-/-}$ cells.

Surprisingly, most of the top upstream regulators predicted by IPA analysis that may govern the observed gene expression changes were shared between *Upf2*$^{-/-}$ and *Pelo*$^{-/-}$ cells and included *Trp53*, *Myc*, *Tgfb1*, *Errb2*, *Cdkn1a*, *Hras*, and *Nfkbia* (*Figure 6G*). In agreement, genes with differential translational efficiency were strongly correlated between *Upf2*$^{-/-}$ and *Pelo*$^{-/-}$ cells (Pearson's correlation, r = 0.6576). Furthermore, differentially translated genes in *Upf2*$^{-/-}$ (32% of genes, DE RPF *Upf2*$^{-/-}$) and *Pelo*$^{-/-}$ (34% of genes, DE RPF *Pelo*$^{-/-}$) cells also showed a strong linear correlation (*Figure 6H*), indicating that defects in these quality control pathways may not only lead to similar changes in translation regulation (TE) but also in global gene translation (DE RPF). Consistent with the similar changes in translation, KEGG pathway analysis of differentially translated genes (DE RPF, adj. p≤0.05) revealed similar enrichment of multiple pathways in *Upf2*$^{-/-}$, *Pelo*$^{-/-}$, and *Hbs1*$^{-/-}$ cells (*Figure 6I*).

In contrast with the many ribosome pauses we observed specifically in *Pelo*$^{-/-}$ cells, only 4% of all ribosome pausing events observed in *Upf2*$^{-/-}$ and control cells were uniquely found in *Upf2*$^{-/-}$ cells (*Figure 6—figure supplement 1D*). Although only a small fraction (8%) of *Pelo-* and *Hbs1l*-specific pausing transcripts corresponded to NMD transcripts, we considered the possibility that some of the protein-coding transcripts with ribosome pauses in *Pelo*$^{-/-}$ and *Hbs1l*$^{-/-}$ cells may also be NMD sensitive given that 5–10% of mRNAs without premature termination codons are thought to be degraded by the NMD pathway (*He et al., 2003*; *Jaffrey and Wilkinson, 2018*; *Lelivelt and Culbertson, 1999*; *Mendell et al., 2004*). However, upregulation of these protein-coding pausing transcripts was not observed in *Upf2*$^{-/-}$ cells (*Figure 6—figure supplement 1E*), suggesting that these pausing transcripts are not NMD sensitive. Together, these findings suggest that while *Pelo/Hbs1l* and *Upf2* largely function in distinct quality control pathways, disruption of either pathway results in similar translational gene expression changes.

## Deletion of *Upf2* or *Pelo* cause similar cerebellar developmental defects

Intrigued by the similar changes in translation upon impairment of *Upf2* and *Pelo/Hbs1l* in MEFs, we conditionally deleted *Upf2* during cerebellar development to determine if phenotypic similarities exist upon loss of these different translation-dependent quality control pathways. Surprisingly, deletion of *Upf2* (*En1*$^{Cre}$; *Upf2* cKO) or *Pelo* (*En1*$^{Cre}$; *Pelo* cKO) using *En1*$^{Cre}$ resulted in largely indistinguishable defects with a grossly hypoplastic cerebellum and regions of the midbrain (superior and

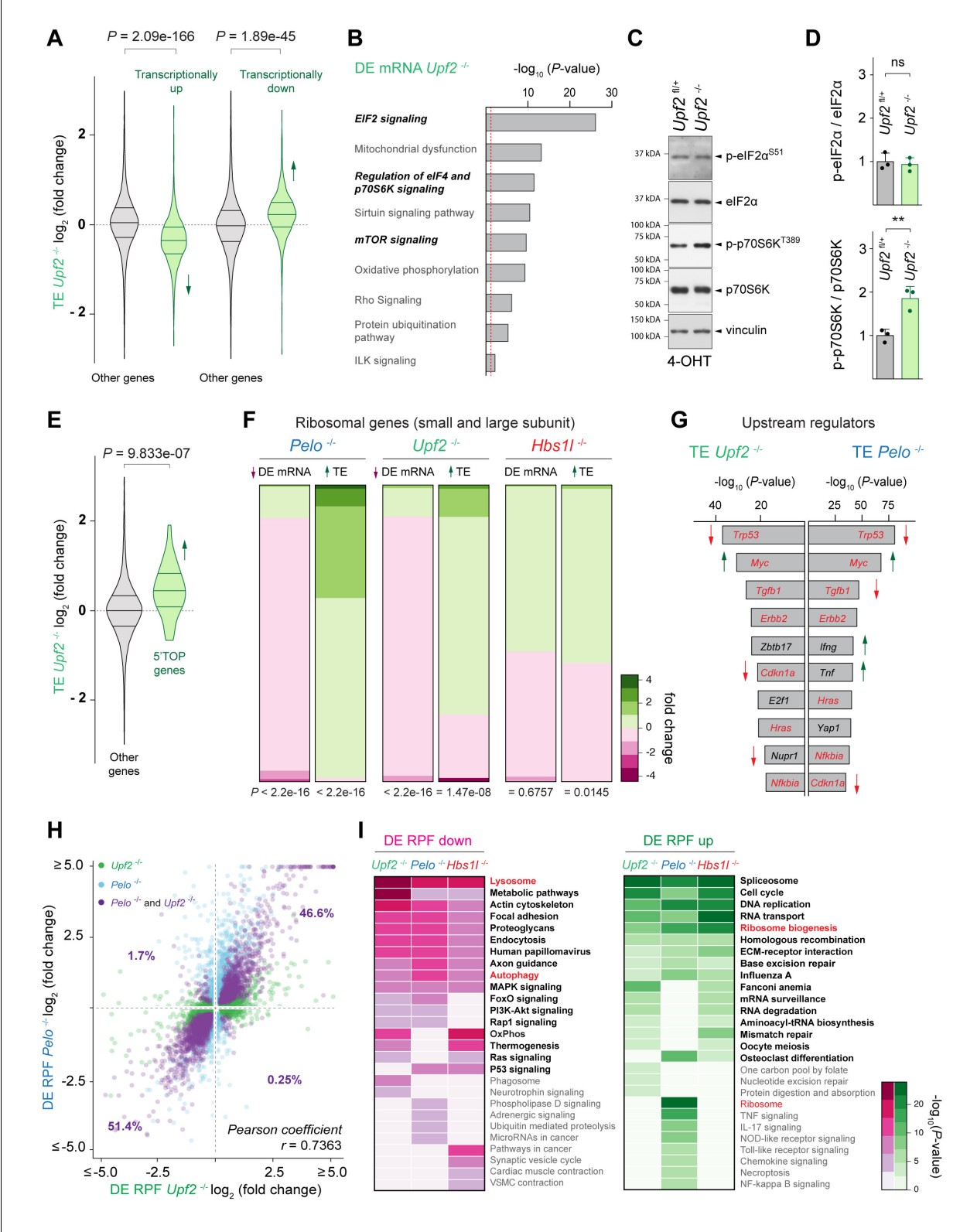

**Figure 6.** Defects in different translation-dependent quality control pathways similarly alter translation regulation and the translatome. (**A**) Translational efficiency (TE) of genes that are transcriptionally upregulated or downregulated was compared to the remaining ('other') genes from *Upf2⁻/⁻* (green) MEFs. Downward direction of the arrow indicates significant decrease in translational efficiency of transcriptionally upregulated genes in *Upf2⁻/⁻* MEFs. Upward direction of the arrow indicates significant increase in translational efficiency of transcriptionally downregulated genes in *Upf2⁻/⁻* MEFs. (**B**)

*Figure 6 continued on next page*

Figure 6 continued

Ingenuity pathway analysis (IPA) of differentially transcribed genes from $Upf2^{-/-}$ MEFs (DE mRNA $Upf2^{-/-}$). EIF2 and mTOR/p70S6K signaling are in italics. The red dashed line indicates the significance threshold (p=0.05). (C) Western blot analysis of p-eIF2α$^{S51}$ and p-p70S6K$^{T389}$ of lysates from tamoxifen-treated control ($Upf2^{fl/+}$) and $Upf2^{-/-}$ MEFs at Day 2 (Passage P1). Vinculin was used as a loading control. (D) Levels of p-eIF2α$^{S51}$ or p-p70S6K$^{T389}$ were normalized to total level of eIF2α or p70S6K, and phosphorylation levels are relative to those of control ($Upf2^{fl/+}$) MEFs. Data represent mean + SD. (E) Translational efficiency (TE) of genes those translation is regulated by mTOR via their 5'TOP motif was compared to the remaining ('other') genes from $Upf2^{-/-}$ (green) MEFs. Upward direction of the arrow indicates significant increase in translational efficiency of 5'TOP genes. (F) Differential transcription (DE mRNA) or translational efficiency (TE) of ribosomal protein genes (small and large ribosomal subunit) was compared to the remaining ('other') genes from $Pelo^{-/-}$ (blue), $Upf2^{-/-}$ (green), and $Hbs1l^{-/-}$ (red) MEFs. Up- and downward direction of arrows indicates significant up- and downregulation of ribosomal protein genes, respectively. The heatmap indicates the gene expression changes of ribosomal protein genes. (G) Identification of upstream regulators of genes with differential translational efficiency in $Upf2^{-/-}$ (TE $Upf2^{-/-}$) and $Pelo^{-/-}$ (TE $Pelo^{-/-}$) MEFs. Top ten transcription factors are shown. Those enriched in both $Upf2^{-/-}$ and $Pelo^{-/-}$ MEFs and shown in red. Up- or downward direction of arrows indicates predicted up- or downregulation of transcription factors, respectively. (H) Differentially translated genes in $Upf2^{-/-}$ MEFs (DE RPF $Upf2^{-/-}$, adj. p≤0.05, x-axis, green) were plotted against genes that are differentially translated in $Pelo^{-/-}$ MEFs (DE RPF $Pelo^{-/-}$, adj. p≤0.05, y-axis, blue). Genes those translation was significantly different in both $Upf2^{-/-}$ and $Pelo^{-/-}$ MEFs are shown in purple. (I) KEGG pathway analysis of differentially translated (up- and downregulated) genes (DE RPF, adj. p≤0.05) in $Upf2^{-/-}$ (green), $Pelo^{-/-}$ (blue) and $Hbs1l^{-/-}$ (red) MEFs. Significantly (p≤0.05) enriched pathways are shown and pathways in bold indicate pathways that are shared between any of the mutant MEFs. Pathways known to be positively or negatively regulated by mTORC1 are in red. 4-OHT, 4-hydroxytamoxifin; 5'TOP, 5'terminal oligopyrimidine motif. Student's t-test (D); Wilcoxon rank-sum test was used to determine statistical significance (A, E, F); Pearson coefficient (r) was determined to analyze linearity of gene expression changes (H). ns, not significant; **p≤0.01.

The online version of this article includes the following source data and figure supplement(s) for figure 6:

**Source data 1.** Defects in different translation-dependent quality control pathways similarly alter translation regulation and the translatome.

**Figure supplement 1.** Loss of $Upf2$ leads to an increase of NMD targets.

---

inferior colliculus) being nearly absent unlike in $En1^{Cre}$; $Hbs1l$ cKO mice (**Figures 7A** and **2D**). Both $Upf2$ and $Pelo$ mutant pups died shortly after birth, perhaps due to $En1^{Cre}$ deletion of these genes in other cell types (**Britz et al., 2015**; **Kimmel et al., 2000**; **Sapir et al., 2004**; **Sgaier et al., 2007**; **Wurst et al., 1994**).

Similar neurogenesis defects were observed in the $Upf2$ and $Pelo$ mutant cerebellum. The fraction of ventricular zone (VZ) progenitors that remained in the cell cycle was higher in E13.5 $Upf2$- or $Pelo$-deficient cerebella compared to controls (**Figure 7—figure supplement 1A and B**), indicating that like $Hbs1l$, loss of $Upf2$ or $Pelo$ impairs cell cycle exit of VZ-progenitors. Inversely, the number of VZ-derived precursors, for example Purkinje cells (Lhx1/5$^+$ cells) and Pax2$^+$ interneurons were reduced by ~83% and~90% in the $En1^{Cre}$; $Upf2$ cKO and $En1^{Cre}$; $Pelo$ cKO cerebellum (**Figure 7—figure supplement 1C,D,E and F**). Glutamatergic deep cerebellar neurons (Tbr1$^+$) were nearly absent in the E13.5 $En1^{Cre}$; $Upf2$ cKO and $En1^{Cre}$; $Pelo$ cKO cerebellum (**Figure 7—figure supplement 1G and H**). In addition, the EGL was missing in $En1^{Cre}$; $Upf2$ cKO and $En1^{Cre}$; $Pelo$ cKO cerebella at both E13.5 and P0 (**Figure 7A**, **Figure 7—figure supplement 1I**), indicating that neurogenesis defects are particularly more severe in the absence of $Upf2$ or $Pelo$ relative to those observed upon $Hbs1l$ loss.

Thus, we considered that in addition to cell cycle exit abnormalities, cell death might also contribute to impaired cerebellar development. Indeed, the number of apoptotic cells (Casp3$^+$ cells) was significantly increased in the $En1^{Cre}$; $Upf2$ cKO and $En1^{Cre}$; $Pelo$ cKO cerebellum compared to that of controls or $En1^{Cre}$; $Hbs1l$ cKO embryos, and apoptotic cells were observed in both the ventricular zone and the prospective white matter where progenitor-derived progeny reside (**Figure 7—figure supplement 1J and K**). Together, these data suggest that defects in cell cycle exit and increased cell death likely impair cerebellar development in $Upf2$ and $Pelo$ mutant mice. To determine if loss of $Upf2$ or $Pelo$ also increased mTORC1 signaling in the developing cerebellum as it did in MEFs, we analyzed levels of p-S6$^{S240/244}$, a known downstream target of mTORC1. In agreement, levels of p-S6$^{S240/244}$ were significantly increased in the $En1^{Cre}$; $Upf2$ cKO and $En1^{Cre}$; $Pelo$ cKO cerebellum, including the VZ (**Figure 7—figure supplement 1L and M**).

Because loss of $Upf2$ or $Pelo$ severely impaired early granule cell neurogenesis, we investigated if these genes are also required later in the development of these cells. Conditional deletion of either $Upf2$ or $Pelo$ using Tg(Atoh1-Cre) resulted in abnormalities of the anterior lobes of the cerebellum in P21 mice (**Figure 7B**). These lobes were reduced in length and Purkinje cells failed to form a monolayer. In contrast, the posterior lobes appeared unaffected, possibly due to the anterior-to-posterior

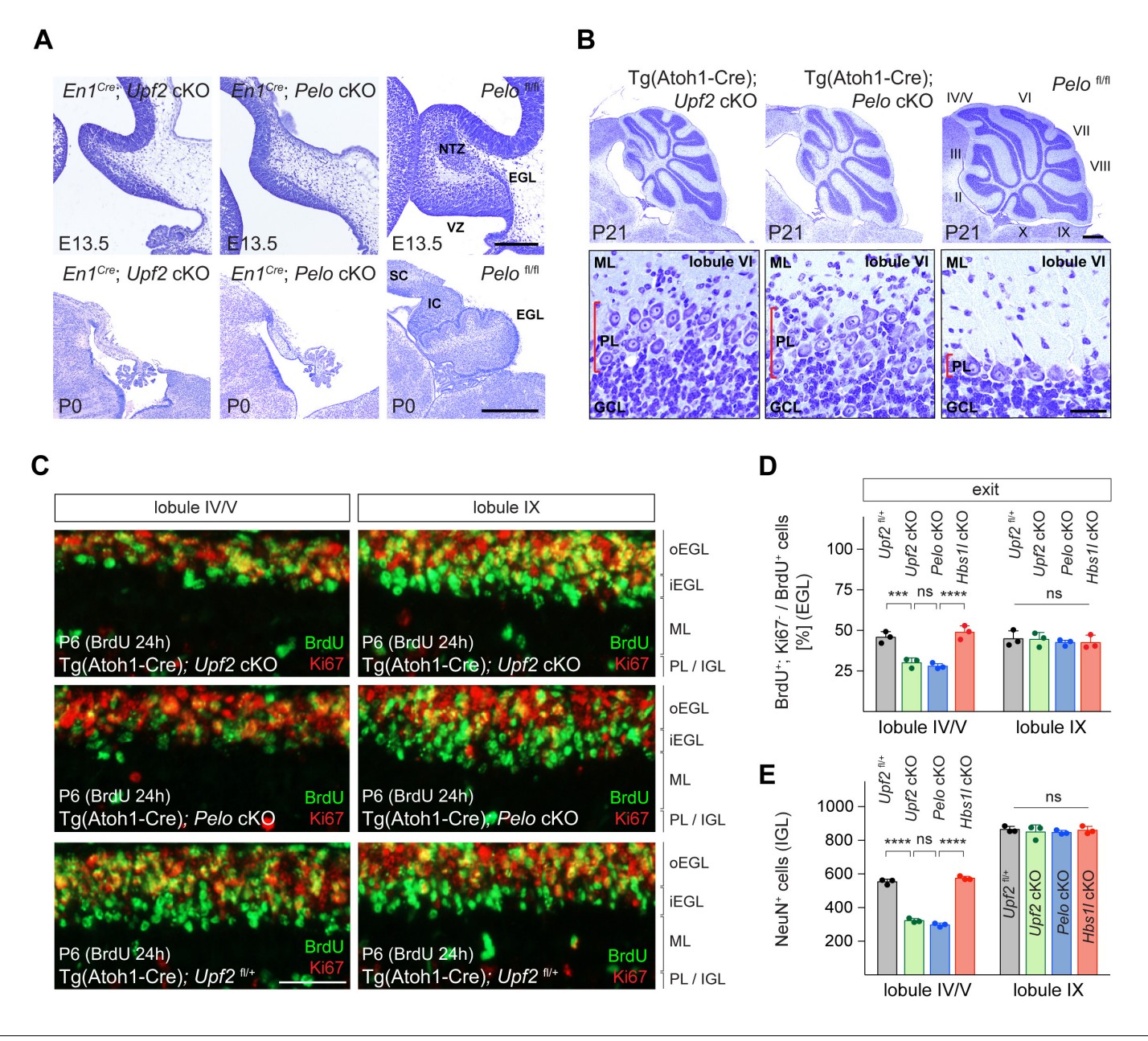

**Figure 7.** Loss of *Upf2* and *Pelo* cause similar cerebellar developmental defects. (**A**) Parasagittal (E13.5) and sagittal (P0) cerebellar sections of control (*Pelo*<sup>fl/fl</sup>), *En1*<sup>Cre</sup>; *Upf2* cKO and *En1*<sup>Cre</sup>; *Pelo* cKO mice stained with cresyl violet. (**B**) Sagittal cerebellar sections of P21 control (*Pelo*<sup>fl/fl</sup>), Tg(Atoh1-Cre); *Upf2* cKO and Tg(Atoh1-Cre); *Pelo* cKO mice stained with cresyl violet. Higher magnification images of lobule VI are shown below for each genotype. Cerebellar lobules are indicated by Roman numerals. (**C**) Immunofluorescence with antibodies to BrdU (green) and Ki67 (red) on sections of P6 control (Tg(Atoh1-Cre); *Upf2*<sup>fl/+</sup>), Tg(Atoh1-Cre); *Upf2* cKO and Tg(Atoh1-Cre); *Pelo* cKO cerebellum. Mice were injected with BrdU 24 hr prior to harvest to determine the fraction of granule cell precursors in the EGL that exited the cell cycle. Images are shown for anterior (IV/V) and posterior (IX) lobules. (**D**) Quantification of the fraction of granule cell precursors in the EGL (lobules IV/V and IX) that exited the cell cycle (BrdU$^+$, Ki67$^-$ cells) of control (Tg (Atoh1-Cre); *Upf2*<sup>fl/+</sup>), Tg(Atoh1-Cre); *Upf2* cKO, Tg(Atoh1-Cre); *Pelo* cKO and Tg(Atoh1-Cre); *Hbs1l* cKO mice. Data represent mean + SD. (**E**) Quantification of terminally differentiated granule cells (NeuN$^+$ cells) in the IGL in lobules IV/V and IX of control (Tg(Atoh1-Cre); *Upf2*<sup>fl/+</sup>), Tg(Atoh1-Cre); *Upf2* cKO, Tg(Atoh1-Cre); *Pelo* cKO and Tg(Atoh1-Cre); *Hbs1l* cKO mice. Data represent mean + SD. Scale bars: 200 µm (E13.5) and 500 µm (P0) (**A**); 500 µm and 20 µm (higher magnification) (**B**); 50 µm (**C**). VZ, ventricular zone, EGL, external granule cell layer; NTZ, nuclear transitory zone; SC, superior colliculus; IC, inferior colliculus; ML, molecular cell layer; PL, Purkinje cell layer; GCL, granule cell layer; oEGL, outer external granule cell layer; iEGL, inner external granule cell layer; IGL, internal granule cell layer. Two-way ANOVA was corrected for multiple comparisons using Tukey method (**D**, **E**). ns, not significant; ***p≤0.001; ****p≤0.0001.

*Figure 7 continued on next page*

*Figure 7 continued*

The online version of this article includes the following source data and figure supplement(s) for figure 7:

**Source data 1.** Loss of *Upf2* and *Pelo* cause similar cerebellar developmental defects.
**Figure supplement 1.** *Upf2* and *Pelo* are required for the development of multiple cerebellar linages.
**Figure supplement 1—source data 1.** *Upf2* and *Pelo* are required for the development of multiple cerebellar linages.

gradient of *Cre* expression in granule cell precursors in the developing cerebellum (*Pan et al., 2009*; *Qiu et al., 2010*; *Wojcinski et al., 2019*).

During postnatal cerebellar development, granule cell precursors in the outer region of the EGL (oEGL) exit the cell cycle and transiently reside in the inner EGL (iEGL) before migrating to the internal granule cell layer (IGL). To determine if granule cell precursors in postnatal Tg(Atoh1-Cre); *Upf2* cKO or Tg(Atoh1-Cre); *Pelo* cKO cerebellum properly exited the cell cycle, we labeled granule cell precursors with BrdU to determine the fraction of precursors that exited the cell cycle twenty-four hours later (BrdU$^+$; Ki67$^-$ cells). In P6 control cerebella, BrdU-labeled granule cell precursors that were negative for Ki67 formed a distinct layer on the ventral surface of the EGL consistent with the appearance and location of the iEGL (*Legué et al., 2016*; *Figure 7C*). However, no clear separation between the oEGL and iEGL was observed in the anterior lobes in Tg(Atoh1-Cre); *Upf2* cKO or Tg (Atoh1-Cre); *Pelo* cKO cerebella (*Figure 7C*), and the fraction of granule cell precursors that exited the cell cycle (BrdU$^+$; Ki67$^-$ cells) in these lobes, but not posterior lobes, was significantly lower compared to control or Tg(Atoh1-Cre); *Hbs1l* cKO cerebella (*Figure 7D*). Correspondingly, the number of terminally differentiated granule cells in the IGL (NeuN$^+$ cells) of the anterior, but not posterior, lobes was reduced by ~45% in P6 Tg(Atoh1-Cre); *Upf2* cKO or Tg(Atoh1-Cre); *Pelo* cKO cerebella (*Figure 7E*), indicating that both of these genes are necessary for differentiation of granule cell precursors.

## Discussion

Translation-dependent quality control pathways govern protein synthesis and proteostasis by degrading aberrant mRNAs that result in potentially toxic peptide products. However, little is known about the in vivo defects or how translation is altered in the absence of these pathways. To interrogate these questions, we investigated the translational and phenotypic alterations from different translation-dependent quality control pathways including the NSD/NGD (*Pelo*/*Hbs1l*) and the NMD (*Upf2*) pathways.

Here, we show that *Pelo* and *Hbs1l* are both critical for mouse embryonic and cerebellar development but dispensable for granule cells after cerebellar development. In addition, loss of *Pelo* or *Hbs1l* locally increase the ribosome occupancy ('ribosome pauses') of genes. Interestingly, the number and strength of these pauses were higher in E8.5 *Hbs1l$^{-/-}$* embryos, just prior to the time when these embryos cease developing, compared to the postnatal cerebellum, where loss of *Hbs1l* had no effect on cerebellar granule cells. Together these data suggest that this translation-dependent quality control pathway may be needed in cell- and/or tissue-specific manner.

In agreement with the differential severity between *Pelo$^{-/-}$*- and *Hbs1l$^{-/-}$*-mediated defects during embryonic and cerebellar development, pauses were stronger and more frequent in *Pelo$^{-/-}$* MEFs and coincided with greater changes in gene expression than in *Hbs1l$^{-/-}$* MEFs. Loss of *Pelo* induced greater translation elongation defects compared to its binding partner *Hbs1l*, which may be in part be influenced by differences in gene expression and translational efficiency between *Pelo*- and *Hbs1l*-deficient cells. However, additional factors may also play a role. For example, while deletion of either *Pelo* or *Hbs1l* led to a reduction of their respective binding partners, the kinetics of this decrease upon deletion of *Pelo* or *Hbs1l* is unknown and may not be equal. In addition, Dom34 (*Pelo*) promotes dissociation of ribosomes by Rli1 (*Abce1*) and this activity increased in the presence of Hbs1 (*Hbs1l*) (*Pisareva et al., 2011*; *Shoemaker and Green, 2011*). Hence, we cannot rule out a scenario in which the remaining levels of PELO in *Hbs1l$^{-/-}$* cells might be able to mitigate and/or delay elongation defects even in the absence of *Hbs1l*.

Most of the ribosome pauses in *Pelo*- and *Hbs1l*-deficient tissues or cells were located in the coding sequence. Although these pauses with a footprint length of 27–34 nucleotides could indicate

ribosomes that paused during translation, phenotypes in *Hbs1l* and *Pelo* mutant mice were not dependent on the B6J-associated mutation in *n-Tr20* that introduces genome-wide AGA pausing within mRNAs. Instead, these pauses may reflect pausing of the 'trailing' ribosomes upstream of the 'lead' ribosome that reached the 3'end of truncated mRNAs (*Guydosh et al., 2017*; *Guydosh and Green, 2014*). Footprints of the 'lead' ribosome have been analyzed on an exosome-deficient background (*Arribere and Fire, 2018*; *D'Orazio et al., 2019*; *Glover et al., 2020*; *Guydosh and Green, 2014*), and their short length of 15–18 nucleotides makes it difficult to uniquely map these footprints in mammals with their larger genomes. Regardless, recent biochemical studies have demonstrated that the *Pelo/Hbs1l* complex rescues trapped ribosomes near the 3'end of truncated mRNAs but is not necessary for the resolution of internally stalled ribosomes within the mRNA (*Juszkiewicz et al., 2020*), which is consistent with early biochemical and structural studies demonstrating that Dom34: Hbs1 preferentially rescues ribosomes arrested at sites of truncation (*Hilal et al., 2016*; *Pisareva et al., 2011*; *Shoemaker and Green, 2011*).

RNA intermediates that converge on *Pelo/Hbs1l* may derive from endonucleolytic cleavage for example during NMD, RNAi, or NGD (*Arribere and Fire, 2018*; *D'Orazio et al., 2019*; *Doma and Parker, 2006*; *Eberle et al., 2009*; *Hashimoto et al., 2017*). However, most of the *Pelo*$^{-/-}$- and *Hbs1l*$^{-/-}$-pausing transcripts were not NMD sensitive, suggesting that if pausing occurred on truncated mRNAs, many of the RNA intermediates likely derived from additional mechanisms. Endonucleolytic cleavage of mRNAs may occur upon persistent ribosome collision during NGD (*D'Orazio et al., 2019*; *Doma and Parker, 2006*; *Shoemaker and Green, 2012*). Intriguingly, widespread ribosome collision has been observed under normal conditions and may represent 10% of the pool of translating ribosomes (*Arpat et al., 2020*; *Diament et al., 2018*; *Han et al., 2020*; *Meydan and Guydosh, 2020*). Numerous human mRNAs are subject to repeated, co-translational endonucleolytic cleavage and occur independent of NMD-associated nucleases (*Ibrahim et al., 2018*). Because translation changes during development and varies between cell types (*Blair et al., 2017*; *Buszczak et al., 2014*; *Castelo-Szekely et al., 2017*; *Gonzalez et al., 2014*; *Sudmant et al., 2018*), RNA intermediates that are generated during translation may also vary and thereby, could introduce a need for quality control pathways in a tissue- and/or cell-type-specific manner.

We also observed that loss of *Pelo/Hbs1l* was associated with the activation of mTORC1 signaling. mTORC1 activation was previously observed in *Hbs1l* patient derived fibroblasts (*O'Connell et al., 2019*) and upon deletion of *Pelo* during epidermal development (*Liakath-Ali et al., 2018*). Interestingly, inhibition of translation/mTORC1 partially prevented *Pelo*$^{-/-}$-mediated epidermal defects (*Liakath-Ali et al., 2018*). How mTORC1 responds to loss of *Pelo/Hbs1l* is unknown. Multiple mTOR-dependent phosphorylation sites on the surface of the ribosome have been observed, suggesting that mTORC1 and/or mTORC1-associated kinases interact with the ribosomes and might provide a mechanism to detect changes in translation elongation (*Jiang et al., 2016*). However, changes in levels of ribosomal mRNAs and/or impaired ribosomal biogenesis also activate mTORC1 signaling (*Liu et al., 2014*). Indeed, deletion of *Pelo/Hbs1l* led to decreased transcript levels but increased translation of ribosomal genes. Thus, mTORC1 may be activated to compensate for decreases in expression of these genes rather than directly sensing defects in translation elongation. To test whether mTORC1 activation was specific to *Pelo/Hbs1l* deficiency or was generally associated with defects in translation-dependent quality control pathways, we examined MEFs deficient for *Upf2*, an essential component of the NMD pathway. Loss of *Upf2* also led to decreased expression and increased translation of ribosomal genes and mTORC1 activation, supporting that changes in mTORC1 signaling are not a direct consequence of the elongation defects, but likely occurs as a compensatory response.

In general, impairment of either translation-dependent quality control pathway led to strikingly similar alterations in gene expression with *Pelo* and *Upf2* showing the most changes compared to *Hbs1l*. Reminiscent of these changes, loss of *Pelo* or *Upf2* had remarkably similar effects on multiple cerebellar neuronal populations during early embryonic and late postnatal neurogenesis, causing comparable morphological defects in the cerebellum (e.g. inhibition of differentiation and an increase in cell death). Growing evidence highlights the role of mTOR in the decision of stem cells to self-renew or differentiate (*Meng et al., 2018*; *Xiang et al., 2011*). However, increased mTOR activity generally reduces self-renewal and promotes differentiation of neuronal stem/progenitor cells (*Hartman et al., 2013*; *LiCausi and Hartman, 2018*; *Magri et al., 2011*; *Way et al., 2009*), suggesting that other molecular changes contributed to the observed defects in neurogenesis.

The similarities in gene expression and developmental alterations in mice deficient for these translation-dependent quality control pathways suggest a convergence of molecular and cellular pathology. In fact, several changes in gene expression and signaling pathways were altered in the same direction in mutant cells. Thus, phenotypic changes could be due to either a change in a single molecular pathway or interactions of multiple pathways. For example, *Myc* activation was predicted as an upstream regulator of gene expression in both *Pelo*[-/-] and *Upf2*[-/-] but less in *Hbs1l*[-/-] MEFs. *Myc* functions as a switch between proliferation and differentiation during cerebellar development (*Knoepfler et al., 2002*; *Ma et al., 2015*; *Wey et al., 2010*), in which loss of *Myc* allows precursors to exit the cell cycle and undergo differentiation, while maintained *Myc* expression retains cells in the proliferation cycle. Intriguingly, previous studies revealed that depletion of NMD factors inhibited differentiation of embryonic stem cells, which coincided with sustained *Myc* expression and its downregulation released the differentiation blockage in NMD-deficient cells (*Li et al., 2015*).

How different translation-dependent quality control pathways can lead to similar changes in translation and cellular defects is unclear. Recent studies in yeast demonstrated that impairment of these quality control mechanisms (*Hbs1*, *Dom34*, *Upf1*, *Upf2*, *Ski7* and *Ski8*) caused protein aggregation (*Jamar et al., 2018*). Protein misfolding occurred co-translationally on highly translated genes and the aggregated proteins overlapped between the different mutant strains. These data suggest that increased translation and protein aggregation may be common properties among different quality control mutants (*Jamar et al., 2018*). Perhaps, regardless of specific targets of various quality control pathways, defects in protein folding, clearance of defective peptide products, and/or defects in mRNA decay may trigger similar cellular responses leading to similar phenotypes.

# Materials and methods

## Key resources table

| Reagent type (species) or resource | Designation | Source or reference | Identifiers | Additional information |
|---|---|---|---|---|
| Strain, strain background (*mouse*) | *Hbs1l*[GTC] | This study (see Materials and methods) | MMRRC #007694-UCD; RRID:MMRRC007694-UCD | International Gene Trap Consortium, IGTC, cell line ID: XE494 |
| Strain, strain background (*mouse*) | *Hbs1l*[tm1a] (C57BL/6N-A[tm1Brd]*Hbs1l*[tm1a(KOMP)Wtsi]) | *Skarnes et al., 2011* | MMRRC #048037-UCD, RRID:MMRRC048037-UCD | |
| Strain, strain background (*mouse*) | B6N.129S4-Gt(ROSA)26Sor[tm1(FLP1)Dym]/J | The Jackson Laboratory | JAX:016226; RRID:IMSRJAX:016226 | |
| Strain, strain background (*mouse*) | B6N.Cg-Edil3Tg(Sox2-Cre)1Amc/J | The Jackson Laboratory | JAX:014094; RRID:IMSRJAX:014094 | |
| Strain, strain background (*mouse*) | B6.FVB-Tg(EIIa-Cre)C5379Lmgd/J | The Jackson Laboratory | JAX:003724; RRID:IMSRJAX:003724 | |
| Strain, strain background (*mouse*) | En1[tm2(Cre)Wrst]/J | The Jackson Laboratory | JAX:007916; RRID:IMSRJAX:007916 | |
| Strain, strain background (*mouse*) | B6.Cg-Tg(Atoh1-Cre)1Bfri/J | The Jackson Laboratory | JAX:011104; RRID:IMSRJAX:011104 | |
| Strain, strain background (*mouse*) | B6.Cg-Tg(CAG-Cre/Esr1*)5Amc/J | | JAX:004682; RRID:IMSRJAX:004682 | |
| Strain, strain background (*mouse*) | B6.Tg(Gabra6-Cre)B1Lfr | *Fünfschilling and Reichardt, 2002* | N/A | |
| Strain, strain background (*mouse*) | B6J-*Pelo*[fl/fl] | This study (see Materials and methods) | N/A | |
| Strain, strain background (*mouse*) | *Upf2*[fl/fl] | *Weischenfeldt et al., 2008* | N/A | |
| Strain, strain background (*mouse*) | B6J.B6N[n-Tr20] | *Ishimura et al., 2014* | N/A | |

*Continued on next page*

*Continued*

| Reagent type (species) or resource | Designation | Source or reference | Identifiers | Additional information |
|---|---|---|---|---|
| Strain, strain background (*mouse*) | B6J-*n-Tr20*[-/-] | *Ishimura et al., 2016* | N/A | |
| Antibody | Anti-phospho-eIF2alpha[S51] (Rabbit polyclonal) | Cell Signaling Technology | CST #9721; RRID:AB_330951 | WB (1:1000) |
| Antibody | Anti-eIF2alpha (Rabbit polyclonal) | Cell Signaling Technology | CST #9722; RRID:AB_2230924 | WB (1:2000) |
| Antibody | Anti-phospho-p70S6K[T389] (Rabbit monoclonal) | Cell Signaling Technology | CST #9234; RRID:AB_2269803 | WB (1:1000) |
| Antibody | Anti-p70S6K (Rabbit monoclonal) | Cell Signaling Technology | CST #2708; RRID:AB_390722 | WB (1:1000) |
| Antibody | Anti-phospho-S6[S240/244] (Rabbit monoclonal) | Cell Signaling Technology | CST #5364; RRID:AB_10694233 | IF (1:1000) |
| Antibody | Anti-cleaved caspase 3 (Rabbit polyclonal) | Cell Signaling Technology | CST #9661; RRID:AB_2341188 | IF (1:100) |
| Antibody | Anti-BrdU (Mouse monoclonal) | Dako/Agilent | M0744; RRID:AB_10013660 | IF (1:50) |
| Antibody | Anti-Hbs1l (Rabbit polyclonal) | Proteintech | #10359–1-AP; RRID:AB_2114730 | WB (1:1000) |
| Antibody | Anti-Pelo (Rabbit polyclonal) | Proteintech | #10582–1-AP; RRID:AB_2236833 | WB (1:2000) |
| Antibody | Anti-Vinculin (Mouse monoclonal) | Sigma-Aldrich | V9131; RRID:AB_477629 | WB (1:20,000) |
| Antibody | Anti-GAPDH (Rabbit monoclonal) | Cell Signaling Technology | CST #2118; RRID:AB_561053 | WB (1:10,000) |
| Antibody | Anti-Ki67 (Rabbit Polyclonal) | Abcam | ab15580; RRID:AB_443209 | IF (1:100) |
| Antibody | Anti-Olig2 (Rabbit monoclonal) | Abcam | ab109186; RRID:AB_10861310 | IF (1:200) |
| Antibody | Anti-Lhx1/5 (Mouse monoclonal) | DSHB | #4F2-c; RRID:AB_531784 | IF (1:100) |
| Antibody | Anti-NeuN (Mouse monoclonal) | Millipore | MAB377; RRID:AB_2298772 | IF (1:500) |
| Antibody | Anti-Tbr1 (Rabbit polyclonal) | Millipore | AB9616; RRID:AB_2200223 | IF (1:1000) |
| Antibody | Anti-Pax2 (Rabbit polyclonal) | Thermo Fisher Scientific | #71–6000; RRID:AB_2533990 | IF (1:50) |
| Antibody | Anti-pH3 (Rabbit polyclonal) | Upstate, Millipore | #06–570; RRID:AB_310177 | IF (1:1000) |
| Antibody | Anti-PCNA (Mouse monoclonal) | Invitrogen, Thermo Fisher Scientific | MA5-11358; RRID:AB_10982348 | IF (1:100) |
| Chemical compound, drug | DNase I | Worthington | LS002139 | |
| Chemical compound, drug | Bromodeoxyuridine (BrdU) | Sigma-Aldrich | B9285 | |
| Chemical compound, drug | 5-Bromo-4-chloro-3-indolyl-β-D-galactopyranoside (X-Gal) | Sigma-Aldrich | B4252 | |

*Continued on next page*

*Continued*

| Reagent type (species) or resource | Designation | Source or reference | Identifiers | Additional information |
|---|---|---|---|---|
| Chemical compound, drug | Z-4-hydroxytamoxifin (4OHT) | Sigma-Aldrich | H7905 | |
| Commercial assay or kit | RNAscope Multiplex Fluorescent Reagent Kit v2 | Advanced Cell Diagnostics | #323100 | |
| Commercial assay or kit | KAPA Stranded mRNA-seq. Kit | Roche | KR0960 | |
| Commercial assay or kit | iQ SYBR Green Supermix | Bio-Rad | #1708880 | |
| Commercial assay or kit | SuperScript III First-Strand Synthesis System | Invitrogen | #18080051 | |
| Commercial assay or kit | DNA-free DNA Removal Kit | Life Technologies | AM1906 | |
| Commercial assay or kit | TSA Plus Cyanine 3 | PerkinElmer | NEL744001KT | |
| Other | RNA-seq. and ribosome profiling data | This study (see Materials and methods) | GSE162556 | Deposited Data |
| Sequence-based reagent | Hbs1l GTC (Genotyping) | This study (see Materials and methods) | N/A | Common Forward:5'AGTCCAGGT GTTTCCTCACG3'; Wild type Reverse:5'CCCTGGCCT ATTTTTGGTTT3'; GTC Reverse:5'TGTCCTCC AGTCTCCTCCAC3' |
| Sequence-based reagent | Hbs1l cKO (Genotyping) | This study (see Materials and methods) | N/A | Forward I: 5'CATGGCCT CCTATGGGTTGA3'; Forward II: 5'GCCTACA GTGAGCACAGAGT3'; Reverse: 5'TAGGTGCTG GGATTTGAACC3' |
| Sequence-based reagent | Pelo cKO (Genotyping) | This study (see Materials and methods) | N/A | Forward:5'TGTAACT GAACCCTGCAGTATCT3'; Reverse I: 5'GTGGAGCATGAAA TGAAATTCGG3'; Reverse II: 5'ATCCAA GGCTTTTACTTCGCC3' |
| Sequence-based reagent | RNAscope probe *Hbs1l*-C2 | Advanced Cell Diagnostics | #527471-C2 | |
| Sequence-based reagent | *Hbs1l* Exon 3–6 (RT-PCR) | This study (see Materials and methods) | N/A | Forward Primer:5'GAAATTGACC AAGCTCGCCTGTA3'; Reverse Primer:5'CTCAGAAGTT AAGCCAGGCACT3' |
| Sequence-based reagent | *β-actin* (RT-PCR) | *Terrey et al., 2020* | N/A | Forward Primer:5'GGCTGT ATTCCCCTCCATCG3'; Reverse Primer:5'CCAGTTGG TAACAATGCCATGT3' |
| Sequence-based reagent | *Hbs1l* Isoform I (quantitative RT-PCR) | This study (see Materials and methods) | N/A | Forward Primer:5'AGACCAT GGGATTTGAAGTGC3'; Reverse Primer:5'CCGGTCT CAGGAATGTTAGGA3' |

*Continued on next page*

*Continued*

| Reagent type (species) or resource | Designation | Source or reference | Identifiers | Additional information |
|---|---|---|---|---|
| Sequence-based reagent | *Hbs1l* Isoform II (quantitative RT-PCR) | This study (see Materials and methods) | N/A | Forward Primer:5'TGAAGTTGAA CAAAGTGCCAAG 3'; Reverse Primer:5'CTGCTTC CTCTGTGTTCCTC3' |
| Sequence-based reagent | *Pelo* (quantitative RT-PCR) | This study (see Materials and methods) | N/A | Forward Primer:5'CCCCAGG AAACGGAAAGGC3'; Reverse Primer:5'ACGCACTTTA CAACCTCGAAG3' |
| Sequence-based reagent | *Gapdh* (quantitative RT-PCR) | *Ishimura et al., 2016* | N/A | Forward Primer: 5'CATTGTCA TACCAGGAAATG3'; Reverse Primer: 5'GGAGAAACC TGCCAAGTATG3' |
| Software, algorithm | Image J | NIH | RRID:SCR_003070; https://imagej.nih.gov/ij | |
| Software, algorithm | GraphPad Prism 7 | GraphPad Prism | RRID:SCR_002798 | |
| Software, algorithm | Pause site identification algorithm | *Ishimura et al., 2014* | N/A | |
| Software, algorithm | biomaRt version 2.42.1 | *Durinck et al., 2005* | RRID:SCR_019214; https://bioconductor.org/packages/release/bioc/html/biomaRt.html | |
| Software, algorithm | ShinyGO v0.61 (KEGG pathway) | *Ge et al., 2020* | RRID:SCR_019213; http://bioinformatics.sdstate.edu/go | |
| Software, algorithm | Ingenuity Pathway Analysis (IPA) | QIAGEN Inc | RRID:SCR_008653; https://www.qiagenbioinformatics.com/products/ingenuity-pathway-analysis | |
| Software, algorithm | DESeq2 v1.26.0 | *Love et al., 2014* | RRID:SCR_015687; https://bioconductor.org/packages/release/bioc/html/DESeq2.html | |
| Software, algorithm | ensembldb v2.6.8 | *Rainer et al., 2019* | RRID:SCR_019103; https://www.bioconductor.org/packages/release/bioc/html/ensembldb.html | |
| Software, algorithm | riborex v2.3.4 | *Li et al., 2017* | RRID:SCR_019104; https://github.com/smithlabcode/riborex | |
| Software, algorithm | RiboWaltz v1.0.1 | *Lauria et al., 2018* | RRID:SCR_016948; https://github.com/LabTranslationalArchitectomics/RiboWaltz | |
| Software, algorithm | bowtie2 v 2.2.3 | *Langmead and Salzberg, 2012* | RRID:SCR_005476; http://bowtie-bio.sourceforge.net/bowtie2/index.shtml | |
| Software, algorithm | fastx_trimmer | Hannon Lab | http://hannonlab.cshl.edu/fastx_toolkit/ | |
| Software, algorithm | fastx_clipper | Hannon Lab | http://hannonlab.cshl.edu/fastx_toolkit/ | |

*Continued*

| Reagent type (species) or resource | Designation | Source or reference | Identifiers | Additional information |
|---|---|---|---|---|
| Software, algorithm | hisat2 v2.1.0 | *Kim et al., 2019* | RRID:SCR_015530; https://daehwankimlab.github.io/hisat2/ | |
| Software, algorithm | featureCounts | *Liao et al., 2014* | RRID:SCR_012919; http://bioinf.wehi.edu.au/featureCounts | |
| Software, algorithm | sleuth v0.30.0 | *Pimentel et al., 2017* | RRID:SCR_016883; https://pachterlab.github.io/sleuth/about | |
| Software, algorithm | kallisto v0.42.4 | *Bray et al., 2016* | RRID:SCR_016582; https://pachterlab.github.io/kallisto/about | |

## Mouse strains

Generation of *Hbs1l*[GTC] mice was performed by injection of targeted ES cells (International Gene Trap Consortium, IGTC, cell line ID: XE494) into C57BL/6J (B6J) blastocysts. B6N-*Hbs1l*[tm1a] (C57BL/6N-A[tm1Brd]*Hbs1l*[tm1a(KOMP)Wtsi], MMRRC:048037) mice were produced at the Wellcome Trust Sanger Institute Mouse Genetics Project as part of the International Mouse Phenotype Consortium (IMPC). In order to generate the conditional *Hbs1l* knock out allele, heterozygous B6N-*Hbs1l*[tm1a/+] mice were crossed to B6N.129S4-Gt(ROSA)26Sor[tm1(FLP1)Dym]/J (The Jackson Laboratory, #016226, MGI:5425632) to remove the gene trap cassette and generate B6N-*Hbs1l*[fl/+] mice. Generation of the constitutive B6N-*Hbs1l*[+/-] knock out allele was accomplished by crossing homozygous B6N-*Hbs1l*[fl/fl] mice to B6N.Cg-Edil3Tg(Sox2-Cre)1Amc/J mice (The Jackson Laboratory, #014094, MGI:4943744). The B6N-*Hbs1l*[+/-] or B6N-*Hbs1l*[fl/fl] mice were backcrossed to congenic B6N.B6J[n-Tr20] mice (n = 2 backcross generations) to introduce the B6J-associated *n-Tr20* mutation. The conditional knock out *Pelo* allele was generated by placing the 5'loxP site 117 bp upstream of exon 2 and the 3'loxP site 302 bp downstream of exon2. Targeted B6J ES cells were injected into B6J blastocysts to generate heterozygous B6J-*Pelo*[fl/+] mice. Generation of the ubiquitous B6J-*Pelo*[+/-] knock out allele was accomplished by crossing homozygous B6J-*Pelo*[fl/fl] mice to B6.FVB-Tg(EIIa-Cre)C5379Lmgd/J mice (The Jackson Laboratory, #003724, MGI:2174520). The conditional *Upf2* knock out allele (*Upf2*[fl/fl] mice) was kindly provided from Drs. Bo Torben Porse and Miles Wilkinson. Genotyping primers are listed below and for the conditional knock out alleles of *Hbs1l* and *Pelo*, genotyping primers were multiplexed to simultaneously detect the wild type, flox (fl) and delta (-) allele. Genotyping for the conditional knock out allele of *Upf2*, was performed as previously described (*Weischenfeldt et al., 2008*).

> *Hbs1l common* Forward: 5'AGTCCAGGTGTTTCCTCACG3'
> *Hbs1l wild type* Reverse: 5'CCCTGGCCTATTTTTGGTTT3'
> *Hbs1l GTC* Reverse: 5'TGTCCTCCAGTCTCCTCCAC3'
> *Hbs1l cKO* Forward I: 5'CATGGCCTCCTATGGGTTGA3'
> *Hbs1l cKO* Forward II: 5'GCCTACAGTGAGCACAGAGT3'
> *Hbs1l cKO* Reverse: 5'TAGGTGCTGGGATTTGAACC3'
> *Pelo cKO* Forward: 5'TGTAACTGAACCCTGCAGTATCT3'
> *Pelo cKO* Reverse I: 5'GTGGAGCATGAAATGAAATTCGG3'
> *Pelo cKO* Reverse II: 5'ATCCAAGGCTTTTACTTCGCC3'

For conditional knock out experiments, the following *Cre*-lines were used and the *Cre* allele was maternally inherited to generate mutant mice (F₂ generation): En1[tm2(Cre)Wrst]/J (*En1*[Cre], The Jackson Laboratory, #007916, MGI:3815003), B6.Cg-Tg(Atoh1-Cre)1Bfri/J (Tg(Atoh1-Cre), The Jackson Laboratory, #011104, MGI:4415810) and B6.Tg(Gabra6-Cre)B1Lfr (Tg(Gabra6-Cre), MGI:4358481, *Fünfschilling and Reichardt, 2002*). In order to avoid the introduction of the B6J-associated *n-Tr20* mutation in *En1*[Cre]; *Hbs1l* cKO mice, En1[tm2(Cre)Wrst]/J mice were backcrossed to congenic B6J.B6N[n-Tr20] mice (*Ishimura et al., 2016*) to generate En1[tm2(Cre)Wrst]/J mice that no longer carry the mutation. Subsequently, B6J.B6N[n-Tr20]; *En1*[Cre] mice were intercrossed with B6N-*Hbs1l*[+/-] to produce F1 mice,

then these mice were crossed to B6N-*Hbs1l*<sup>fl/fl</sup> mice to generate *En1*<sup>*Cre*</sup>; *Hbs1l* cKO; *n-Tr20*<sup>B6N/B6N</sup> without the B6J-associated tRNA mutation. To generate Tg(Atoh1-Cre); *Hbs1l* cKO mice lacking *n-Tr20*, B6.Cg-Tg(Atoh1-Cre)1Bfri/J, B6N-*Hbs1l*<sup>+/-</sup>, and B6N-*Hbs1l*<sup>fl/fl</sup> mice were crossed to B6J-*n-Tr20*<sup>-/-</sup> mice. Subsequently, these strains were intercrossed to generate Tg(Atoh1-Cre); *Hbs1l* cKO; *n-Tr20*<sup>-/-</sup> mice. For BrdU experiments, pregnant females or pups were injected with BrdU (0.05 mg/ g, Sigma-Aldrich, B9285) and collected 30 min (S-phase analysis) or 24 hr (cell cycle exit analysis) post injection. For the isolation of MEFs or embryos, the day that a vaginal plug was detected was defined as embryonic day 0.5 (E0.5).

For conditional knock out experiments in primary mouse embryonic fibroblasts (MEFs), we crossed mice to the tamoxifen inducible *Cre*-line B6.Cg-Tg(CAG-Cre/Esr1*)5Amc/J (CAG-Cre<sup>ER</sup>, The Jackson Laboratory, #004682, MGI:2680708). The *Cre* allele was paternally inherited to generate embryos (F<sub>2</sub> generation) because *Cre*-mediated recombination ('leaky Cre expression') was occasionally observed even in the absence of tamoxifen (4-OHT) treatment of MEFs when the *Cre* allele was maternally inherited (F<sub>2</sub> generation).

All experiments and quantifications were performed with at least three mice/embryos of each genotype and time point using mice of either sex (embryos were not sexed). The Jackson Laboratory Animal Care and Use Committee and The University of California San Diego Animal Care and Use Committee approved all mouse protocols.

### Strain abbreviation

For conditional knock out (cKO) experiments, animals were given a simplified abbreviation throughout the article. The complete genotype is shown below.

| Abbreviation | Genotype |
|---|---|
| *En1*<sup>*Cre*</sup>, *Hbs1l* cKO | *En1*<sup>*Cre*/+</sup>; *Hbs1l*<sup>fl/-</sup>; *n-Tr20*<sup>B6J/B6J</sup> |
| *En1*<sup>*Cre*</sup>, *Pelo* cKO | *En1*<sup>*Cre*/+</sup>; *Pelo*<sup>fl/fl</sup>; *n-Tr20*<sup>B6J/B6J</sup> |
| *En1*<sup>*Cre*</sup>, *Upf2* cKO | *En1*<sup>*Cre*/+</sup>; *Upf2*<sup>fl/fl</sup>; *n-Tr20*<sup>B6J/B6J</sup> |
| Tg(Atoh1-Cre), *Hbs1l* cKO | Tg(Atoh1-Cre); *Hbs1l*<sup>fl/-</sup>; *n-Tr20*<sup>B6J/B6J</sup> |
| Tg(Atoh1-Cre), *Pelo* cKO | Tg(Atoh1-Cre); *Pelo*<sup>fl/fl</sup>; *n-Tr20*<sup>B6J/B6J</sup> |
| Tg(Atoh1-Cre), *Upf2* cKO | Tg(Atoh1-Cre); *Upf2*<sup>fl/fl</sup>; *n-Tr20*<sup>B6J/B6J</sup> |
| Tg(Gabra6-Cre), *Hbs1l* cKO | Tg(Gabra6-Cre); *Hbs1l*<sup>fl/-</sup>; *n-Tr20*<sup>B6J/B6J</sup> |
| Tg(Gabra6-Cre), *Pelo* cKO | Tg(Gabra6-Cre); *Pelo*<sup>fl/-</sup>; *n-Tr20*<sup>B6J/B6J</sup> |
| TgCAG-Cre<sup>ER</sup> (control MEFs) | Tg(CAG-Cre/Esr1); *n-Tr20*<sup>B6J/B6J</sup> |
| TgCAG-Cre<sup>ER</sup>, *Hbs1l* cKO (*Hbs1l*<sup>-/-</sup> MEFs) | Tg(CAG-Cre/Esr1); *Hbs1l*<sup>fl/-</sup>; *n-Tr20*<sup>B6J/B6J</sup> |
| TgCAG-Cre<sup>ER</sup>, *Pelo* cKO (*Pelo*<sup>-/-</sup> MEFs) | Tg(CAG-Cre/Esr1); *Pelo*<sup>fl/fl</sup>; *n-Tr20*<sup>B6J/B6J</sup> |
| TgCAG-Cre<sup>ER</sup>, *Upf2* cKO (*Upf2*<sup>-/-</sup> MEFs) | Tg(CAG-Cre/Esr1); *Upf2*<sup>fl/fl</sup>; *n-Tr20*<sup>B6J/B6J</sup> |

### Cell culture

Primary mouse embryonic fibroblasts (MEFs) were isolated on embryonic day E13.0 and prepared by standard procedures (*Nagy et al., 2014*). MEFs were maintained in Dulbecco's modified Eagle's medium (Gibco, #41965039) with GlutaMAX (Gibco, #35050061), PSN (Gibco, # 15640055), and 10% embryonic stem cell fetal bovine serum (Gibco, #10439024) at 37°C in 5% $CO_2$. Two days post-isolation (Passage P0), the cell culture media was replaced with fresh media and supplemented with 1 µM 4-OHT (4-hydroxytamoxifen, Sigma, H7904) for both control and mutant cells. After 48 hr, cells were washed, trypsinized, and seeded on a 10 cm dish (Passage P1). For RNA sequencing and ribosome footprint profiling experiments, cells were collected 48 hr (Passage P1, Day 2) later when cells reached ~80% confluency. For western blotting experiments, cells of Passage P1 were collected 48 hr (Day 2) and 96 hr (Day 4) post-seeding.

## Histology and immunofluorescence

Anesthetized mice were transcardially perfused with 4% paraformaldehyde (PFA, for immunofluorescence and histology), Bouins fixative (for histology) or 10% neutral buffered formalin (NBF, for in situ hybridization). Tissues were post-fixed overnight and embedded in paraffin. For histological analysis, sections were deparaffinized, rehydrated, and were stained with cresyl violet according to standard procedures. Histological slides were imaged using a digital slide scanner (Hamamatsu).

All quantifications were performed with three mice or embryos of each genotype and time point using animals of either sex (embryos were not sexed). For cell quantification in embryos at E12.5 to E16.5 and pups at P5 (Olig2$^+$ cells), cells were counted from the entire section and values were averaged from three parasagittal (embryos) or sagittal (pups) sections (spaced 35 µm apart) per mouse. For the analysis of granule cell precursors or granule cells in P5 or P6 pups, the total number of cells was determined within lobule IV/V or IX, and values were averaged from three sagittal sections (spaced 50 µm apart) per mouse. For the analysis of granule cells in adult mice (35 weeks of age), granule cells were counted in a 0.025 mm$^2$ area of lobule IX and values were averaged from three midline sections per mouse spaced 100 µm apart.

For immunofluorescence, antigen retrieval on deparaffinized PFA-fixed sections was performed by microwaving sections in 0.01M sodium citrate buffer (pH 6.0, 0.05% Tween-20) for three times with 3 min each or three times for 3 min, followed by two times for 9 min. PFA-fixed sections were incubated with the following primary antibodies overnight at 4°C: rabbit anti-cleaved caspase 3 (Cell signaling, #9661, 1:100), mouse anti-BrdU (Dako, M0744, 1:50), rabbit anti-Ki67 (Abcam, ab15580, 1:100), mouse anti-Lhx1/5 (DSHB, 4F2-c, 1:100), mouse anti-NeuN (Millipore, MAB377, 1:500), mouse anti-PCNA (Invitrogen, MA5-11358, 1:100), rabbit anti-p-S6$^{240/244}$ (Cell Signaling, #5364, 1:1000), rabbit anti-Pax2 (ThermoFisher, 71–6000, 1:50), rabbit anti-Olig2 (Abcam, ab109186, 1:200), rabbit anti-Tbr1 (Chemicon, AB9616, 1:1000), and rabbit anti-pH3 (Upstate, 06–570, 1:1000). Immunofluorescence with antibodies to BrdU was performed on sections treated with DNase I (5mU/µl, Worthington, LS002139) for 45 min at 37°C after antigen retrieval. Detection of primary antibodies was performed with goat anti-mouse Alexa Fluor-488 or −555, goat anti-rabbit Alexa Fluor-488 or −555, and donkey anti-rabbit Alexa Fluor-555 secondary antibodies (Invitrogen). Sections were counterstained with DAPI and Sudan Black to quench autofluorescence.

For immunofluorescence quantification, the fluorescence intensity was measured in an area of 60 × 125 µm using ImageJ, averaged from three sections (spaced 35 µm apart) per embryo and expressed as the fold change relative to control.

## RNAscope (in situ hybridization)

In situ hybridization of *Hbs1l-C2* probes (ACD, #527471-C2) was performed with the ACD RNAscope Multiplex Fluorescent Reagent Kit v2 (ACD, #323100) using the manufacturer's protocol. Briefly, deparaffinized NBF-fixed sections were treated for 15 min with Target Retrieval Reagent at 100°C and subsequently, treated with Protease Plus for 20 min (E13.5 embryos) or 30 min (P28 mice) at 40°C. RNAScope probes were hybridized for 2 hr; TSA Plus Cyanine 3 (PerkinElmer, 1:1,500) was used as a secondary fluorophore for *Hbs1l-C2* probes.

## Reverse transcription and quantitative PCR (qPCR) analysis

Whole mouse embryos, primary mouse embryonic fibroblasts, or adult mouse tissues were isolated and immediately frozen in liquid nitrogen. Total RNA was extracted with Trizol reagent (Life Technologies). cDNA synthesis was performed on DNase-treated (DNA-free DNA Removal Kit, Life Technologies AM1906) total RNA using oligo(dT) primers and SuperScript III First-Strand Synthesis System (Life Technologies). Quantitative RT-PCR reactions were performed using iQ SYBR Green Supermix (Bio-Rad) and an CFX96 Real-Time PCR Detection System (Bio-Rad). Expression levels of *β-actin* were used as input control for semi-quantitative RT-PCR. For quantitative RT-PCR (qPCR) analysis, expression levels of the genes of interest were normalized to *Gapdh* using the 2$^{-\Delta\Delta CT}$ method (*Livak and Schmittgen, 2001*) and expressed as the fold change + standard error of the mean (SEM) relative to control.

Semi-quantitative RT-PCR Primers (F, Forward; R, Reverse):

*Hbs1l Exon 3* F: 5'GAAATTGACCAAGCTCGCCTGTA3'
*Hbs1l Exon 6* R: 5'CTCAGAAGTTAAGCCAGGCACT3'

β-actin F: 5'GGCTGTATTCCCCTCCATCG3'
β-actin R: 5'CCAGTTGGTAACAATGCCATGT3'

Quantitative RT-PCR Primers (F, Forward; R, Reverse):

*Hbs1l* F: 5'AGACCATGGGATTTGAAGTGC3'
*Hbs1l* R: 5'CCGGTCTCAGGAATGTTAGGA3'
*Hbs1l II* F: 5'TGAAGTTGAACAAAGTGCCAAG3'
*Hbs1l II* R: 5'CTGCTTCCTCTGTGTTCCTC3'
*Pelo* F: 5'CCCCAGGAAACGGAAAGGC3'
*Pelo* R: 5'ACGCACTTTACAACCTCGAAG3'
*Gapdh* F: 5'CATTGTCATACCAGGAAATG3',
*Gapdh* R: 5'GGAGAAACCTGCCAAGTATG3'

## Western blotting

MEFs or tissues were immediately frozen in liquid nitrogen. Proteins were extracted by homogenizing frozen tissue or cell samples in five volumes of RIPA buffer with cOmplete Mini, EDTA-free Protease inhibitor Cocktail (Roche), sonicating tissues two times for 10 s (Branson, 35% amplitude) or triturating cells 10 times using a 26G needle. Lysates were incubated for 30 min at 4°C, centrifuged at 16,000xg for 25 min, and 25 µg of whole protein lysate were resolved on SDS-PAGE gels prior to transfer to PVDF membranes (GE Healthcare Life Sciences, #10600023) using a tank blotting apparatus (BioRad).

For detection of phosphoproteins, frozen tissue samples were homogenized in 5 volumes of homogenization buffer (50 mM Hepes/KOH, pH 7.5, 140 mM potassium acetate, 4 mM magnesium acetate, 2.5 mM dithiothreitol, 0.32M sucrose, 1 mM EDTA, 2 mM EGTA) (*Carnevalli et al., 2004*), supplemented with phosphatase and protease inhibitors (PhosStop and cOmplete Mini, EDTA-free Protease inhibitor Cocktail, Roche). Frozen cell samples were homogenized by using a 26G needle (five times) in homogenization buffer (see above). Lysate samples were immediately centrifuged at 12,000xg for 7 min and whole protein lysate were resolved on SDS-PAGE gels prior to transfer to PVDF membranes.

After blocking in 5% nonfat dry milk (Cell Signaling, #9999S), blots were probed with primary antibodies at 4°C overnight: rabbit anti-phospho-eIF2α$^{S51}$ (Cell Signaling, #9721, 1:1000), rabbit anti-eIF2α (Cell Signaling, #9722, 1:2000), rabbit anti-Hbs1l (Proteintech, 10359–1-AP, 1;1000), rabbit anti-Pelo (Proteintech, 10582–1-AP, 1:2000), rabbit anti-GAPDH (Cell Signaling, #2118, 1:10,000), rabbit anti-phospho-p70S6K$^{T389}$ (Cell Signaling, #9234, 1:1000), rabbit anti-phospho-p70S6K (Cell Signaling, #2708, 1:1000), mouse anti-vinculin (Sigma, V-9131, 1:20,000). Primary antibodies were detected by incubation with HRP-conjugated secondary antibodies for 2 hr at room temperature: goat anti-rabbit IgG (BioRad, #170–6515) or goat anti-mouse IgG (BioRad, #170–6516). Signals were detected with SuperSignal West Pico Chemiluminescent substrate (ThermoScientific, #34080) or Pro-Signal Femto ECL Reagent (Genesee Scientific, #20–302).

## RNA sequencing library construction

For RNA sequencing of primary mouse embryonic fibroblasts (MEFs), one 10 cm dish of tamoxifen (4-OHT) treated MEFs at Day 2 (Passage P1) from TgCAG-Cre$^{ER}$ (control), TgCAG-Cre$^{ER}$; *Hbs1l* cKO (*Hbs1l$^{-/-}$*), TgCAG-Cre$^{ER}$; *Pelo* cKO (*Pelo$^{-/-}$*) and TgCAG-Cre$^{ER}$; *Upf2* cKO (*Upf2$^{-/-}$*) were collected. One 10 cm dish from one individual embryo was used as one biological replicate, and either two (*Pelo*) or three (control, *Hbs1*, *Upf2*) biological replicates were used per genotype. Two micrograms of DNase-treated (DNA-free DNA Removal Kit, Life Technologies AM1906) total RNA were used for the RNA library construction, performed as per the manufacturer's protocol (KAPA Stranded mRNA-Seq Kit, KR0960) and the adapter ligation was performed for 3 hr at room temperature. Library quality and concentration was assessed using D1000 screen tape on the Agilent TapeStation and Qubit 2.0 Fluorometer. All libraries were pooled, and the pool of libraries was sequenced on two lanes using HiSeq4000 (PE100).

## Analysis of RNA sequencing data

Reads were quantified using kallisto version 0.42.4 (*Bray et al., 2016*) and pseudo-aligned to a Gencode M24 transcriptome reference with parameters –bias and -b 100. Differential expression was

performed using sleuth version 0.30.0 (*Pimentel et al., 2017*). Pairwise comparisons were performed to identify differentially expressed transcripts and genes: TgCAG-Cre$^{ER}$ vs. TgCAG-Cre$^{ER}$; *Pelo* cKO, TgCAG-Cre$^{ER}$ vs. TgCAG-Cre$^{ER}$; *Hbs1l* cKO, and TgCAG-Cre$^{ER}$ vs. TgCAG-Cre$^{ER}$; *Upf2* cKO. We used functions within sleuth to perform differential transcript and gene expression. Briefly, we fit null models and models corresponding to the genotype of the samples for each transcript and performed Wald tests on the models for each transcript to identify differentially expressed transcripts. Differential gene expression was performed by aggregating transcript expression on Ensembl gene identifiers. Multiple hypotheses testing was corrected using Benjamini-Hochberg correction, referred to as q-value. Transcript biotypes were identified using biomaRt version 2.42.1 (*Durinck et al., 2005*). For downstream TE analysis, mapping to mm10 using a Gencode M24 transcript annotation was performed using hisat2 version 2.1.0 (*Kim et al., 2019*) using default parameters.

## Ribosome profiling library construction

Ribosome profiling libraries were generated as previously described (*Ingolia et al., 2012*; *Ishimura et al., 2014*) with some modifications. Cerebella were dissected and immediately frozen in liquid nitrogen. One cerebellum from P14 mice was used for each biological replicate, and three biological replicates were prepared for each control (Tg(Atoh1-Cre); *Hbs1l*$^{+/+}$) and mutant (Tg(Atoh1-Cre); *Hbs1l* cKO) genotype. For profiling of mouse embryos, embryos from timed mating were dissected at embryonic day E8.5, frozen in a drop of nuclease free water on dry ice, and then flash frozen in liquid nitrogen. Five embryos were pooled for each biological replicate, and three biological replicates were prepared for each control (*Hbs1l*$^{+/+}$) and mutant (*Hbs1l*$^{-/-}$) genotype. For profiling of primary mouse embryonic fibroblasts (MEFs), tamoxifen (4-OHT) treated MEFs at Day 2 (Passage P1) from TgCAG-Cre$^{ER}$ (control), TgCAG-Cre$^{ER}$; *Hbs1l* cKO (*Hbs1l*$^{-/-}$), TgCAG-Cre$^{ER}$; *Pelo* cKO (*Pelo*$^{-/-}$) and TgCAG-Cre$^{ER}$; *Upf2* cKO (*Upf2*$^{-/-}$) were collected. Two 10 cm dishes from one individual embryo were pooled as one biological replicate, and three biological replicates (three individual embryos) were used for each genotype. The tissue and embryo homogenization were performed with a mixer mill (Retsch MM400) in lysis buffer (20 mM Tris-Cl, pH 8.0, 150 mM NaCl, 5 mM MgCl$_2$, 1 mM DTT, 100 µg/ml CHX, 1% (v/v) TritonX-100, 25units/ml Turbo DNase I). The cell homogenization was performed by triturating the cells in lysis buffer (20 mM Tris-Cl, pH 8.0, 150 mM NaCl, 5 mM MgCl$_2$, 1 mM DTT, 100 µg/ml CHX, 1% (v/v) TritonX-100, 25units/ml Turbo DNase I) ten times through a 26G needle. RNase I-treated lysates were overlaid on top of a sucrose cushion in 5 ml Beckman Ultraclear tubes and centrifuged in an SW55Ti rotor for 4 hr at 4°C at 46,700 rpm. Pellets were resuspended and RNA was extracted using the miRNeasy kit (Qiagen) according to manufacturer's instructions. 26–34 nts (cerebella samples) or 15–34 nts (embryo and MEF samples) RNA fragments were purified by electrophoresis on a denaturing 15% gel. Linker addition, cDNA generation (first-strand synthesis was performed at 50°C for 1 hr), circularization, rRNA depletion, and amplification of cDNAs with indexing primers were performed. Library quality and concentration was assessed using high sensitivity D1000 screen tape on the Agilent TapeStation and Qubit 2.0 Fluorometer. All libraries were pooled as set of six samples consisting of three control and mutant samples. Libraries were run on HiSeq4000 (SR75) and either three lanes (cerebella samples) or two lanes (embryo and MEF samples) per set of samples were sequenced.

## Analysis of ribosome profiling data

Reads were clipped to remove adaptor sequences (CTGTAGGCACCATCAAT) using fastx_clipper and trimmed so that reads start on the second base using fastx_trimmer (http://hannonlab.cshl.edu/fastx_toolkit/). Reads containing ribosomal RNAs, snoRNAs, and tRNAs were then filtered out by mapping to a ribosomal RNA reference using bowtie2 version 2.2.3 using parameters -L 13 (*Langmead and Salzberg, 2012*). Remaining reads were mapped to an mm10 mouse reference using a Gencode M24 annotation, or a Gencode M24 protein-coding transcript reference using hisat2 version 2.1.0 (*Kim et al., 2019*). Ribosomal A-sites were identified using RiboWaltz version 1.0.1 (*Lauria et al., 2018*), and reads with lengths of 27–34 nucleotides were retained for further analysis.

To identify potential codon bias in the A-site, observed/expected reads were calculated for each transcript with alignments with the expected reads being the read density expected at a given site with a given codon, assuming that reads are uniformly distributed across the coding part of the

transcript. Differences in codon usage were then tested with a student's t-test followed by Benjamini-Hochberg multiple hypothesis testing correction.

Pauses were identified using previous methodology (*Ishimura et al., 2014*). Reads with lengths of 27–34 nucleotides were analyzed using a 0.5 reads/codon threshold in all samples to analyze pausing on transcripts. Additionally, pauses with P-sites nearby start and stop codons (P-sites at the −3, 0, 1, 2, 3, and 4 positions for start codons and P-sites at the −1 and −2 positions for stop codons) in any isoform of the gene were excluded from the analysis. To identify pauses overlapping start and stop codons in other isoforms, we used ensembldb v.2.6.8 (*Rainer et al., 2019*) to map transcript coordinates back to the genome and analyzed whether pauses with identical genome coordinates overlapped start/stop codons.

To extend this approach to identify pauses at the gene level, we removed start/stop overlapping pauses (in any gene-matched transcript) and collapsed pauses based on the genomic location of the A-site, which was also identified with ensembldb v.2.6.8. Mean pause score across transcripts of genomically identical pauses for each sample was the reported pause score (excluding transcripts without a reported pause in a sample). If genomically identical pauses were in multiple transcript regions (i.e. 3'UTR or CDS depending on the isoform), all were reported.

Pauses were further filtered such that the pause locus appeared in all three replicates of the genotype and thereby, pauses that were only detected in one or two of three replicates of the genotype were removed from the analysis. Pauses were compared between the 4-OHT-treated knock out (TgCAG-Cre^ER; *Hbs1l* cKO, TgCAG-Cre^ER; *Pelo* cKO and TgCAG-Cre^ER; *Upf2* cKO) and the control (TgCAG-Cre^ER) sample that each mutant sample was pooled with for sequencing.

To identify ribosome footprint reads with 3'end As (untemplated reads), we extracted at first reads from a genome mapped bam file with six or more As at the 3'end of the read (3'end As were soft-clipped). Mapped reads were only considered if the 3'end A reads were not matching the reference sequence, while all unmapped reads with six or more As at the 3' end were considered. Afterwards, the 3'end As were then removed from these untemplated reads and they were mapped back to the transcriptome using parameters described above for ribosome footprint profiling mapping to the transcriptome. Only the untemplated reads that mapped after the removal of the 3'end As were considered as ribosome footprints that may derive from ribosomes translating premature polyadenylated transcripts.

For differential RPF and differential TE analysis, genome mapped reads were quantified using featureCounts (*Liao et al., 2014*) with footprints overlapping CDS features. For TE analysis specifically, RNA-Seq read pairs overlapping gene exon features were also quantified using featureCounts. Differential RPF analysis was performed using DESeq2 (v1.26.0) (*Love et al., 2014*) comparing 4-OHT-treated knock out (TgCAG-Cre^ER; *Hbs1l* cKO, TgCAG-Cre^ER; *Pelo* cKO and TgCAG-Cre^ER; *Upf2* cKO) to control (TgCAG-Cre^ER) cells using default parameters. Histone mRNAs were removed from the analysis by removing gene names with the prefixes 'Hist', 'H1f', 'H2a', 'H2b', 'H3', and 'H4'. TE was quantified and tested for using riborex version 2.3.4 using the DESeq2 engine (*Li et al., 2017*).

## Pathway analysis

Data were analyzed using Ingenuity Pathway Analysis (IPA, QIAGEN Inc, https://www.qiagenbioinformatics.com/products/ingenuity-pathway-analysis).

Kyoto Encyclopedia of Genes and Genomes (KEGG) pathway analysis was performed using the ShinyGO v0.61 bioinformatics web server (http://bioinformatics.sdstate.edu/go) (*Ge et al., 2020*) by uploading the gene lists from our RNA sequencing or ribosome profiling analysis. Pathway enrichment terms with a *P*-value cutoff (FDR) $\leq 0.05$ were considered enriched.

## Statistics

For quantification of protein expression, RNA expression (quantitative RT-PCR), fluorescence intensity or histological quantifications, p-values were computed in GraphPad Prism using either student's t-test, multiple t-tests, one-way ANOVA, or two-way ANOVA and statistical tests were corrected for multiple comparisons as indicated in the figure legends. All quantifications were performed with at least three mice/embryos of each genotype and time point using mice of either sex (embryos were not sexed).

## Acknowledgements

We thank T Jucius and A Kano for technical assistance, Drs. Bo Torben Porse and Miles Wilkinson for providing the *Upf2*<sup>fl/fl</sup> mice, the IGM Genomics Center at the University of California San Diego (UCSD) for support with sequencing and the UCSD School of Medicine Microcopy Core for providing access to microscopy equipment (Grant P30 NS047101). This work was supported in part by NIH R01 NS094637 (SLA). SLA is an investigator of the Howard Hughes Medical Institute.

## Additional information

### Funding

| Funder | Grant reference number | Author |
| --- | --- | --- |
| National Institute of Neurological Disorders and Stroke | NS094637 | Susan L Ackerman |
| Howard Hughes Medical Institute | | Susan L Ackerman |

The funders had no role in study design, data collection and interpretation, or the decision to submit the work for publication.

### Author contributions

Markus Terrey, Conceptualization, Formal analysis, Investigation, Visualization, Methodology, Writing - original draft, Writing - review and editing; Scott I Adamson, Formal analysis, Investigation, Visualization, Methodology, Writing - review and editing; Jeffrey H Chuang, Formal analysis, Supervision, Investigation, Writing - review and editing; Susan L Ackerman, Conceptualization, Supervision, Funding acquisition, Investigation, Methodology, Writing - original draft, Project administration, Writing - review and editing

### Author ORCIDs

Markus Terrey (iD) https://orcid.org/0000-0002-7359-4810
Susan L Ackerman (iD) https://orcid.org/0000-0002-6740-593X

### Ethics

Animal experimentation: Animal Protocol number S15286.

### Decision letter and Author response

Decision letter https://doi.org/10.7554/eLife.66904.sa1
Author response https://doi.org/10.7554/eLife.66904.sa2

## Additional files

### Supplementary files

- Supplementary file 1. Embryonic lethality.
- Supplementary file 2. Locus specific pausing genome level.
- Supplementary file 3. DE RPF footprints.
- Supplementary file 4. A-site pausing.
- Supplementary file 5. Locus-specific pausing transcript level.
- Supplementary file 6. TE MEFs.
- Supplementary file 7. DE mRNA MEFs.
- Transparent reporting form

## Data availability

Sequencing data have been deposited in GEO under accession code GSE162556.

The following dataset was generated:

| Author(s) | Year | Dataset title | Dataset URL | Database and Identifier |
|---|---|---|---|---|
| Ackerman SL, Terrey M, Adamson SI, Chuang JH | 2021 | Mutations in distinct translation-dependent quality control pathways lead to convergent molecular pathogenesis and neurodevelopmental defects. | https://www.ncbi.nlm.nih.gov/geo/query/acc.cgi?acc=GSE162556 | NCBI Gene Expression Omnibus, GSE162556 |

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
