## [Decision Letter]

**Acceptance summary:**

This work addresses the role of translation-dependent quality control pathways in mouse neuronal development. Not only are the ribosome rescue factors Pelo and Hbs1l shown to be required for early embryonic development and for neurogenesis in the developing cerebellum, but the detailed underlying molecular defects in translation elongation are also characterized. In addition, it is shown that inactivation of the nonsense mediated decay factor Upf2 has similar effects on cell cycle exit and cell death of granule cell progenitors as Pelo has. The study is carried out extremely well, demonstrating the importance of translation-dependent quality control, a fundamental cellular process, during neuronal development.

**Decision letter after peer review:**

Thank you for submitting your article "Defects in translation-dependent quality control pathways lead to convergent molecular and neurodevelopmental pathology" for consideration by *eLife*. Your article has been reviewed by 2 peer reviewers, including Mary E Hatten as the Reviewing Editor and Reviewer #1, and the evaluation has been overseen by Huda Zoghbi as the Senior Editor.

Essential Revisions:

1) Please consider the queries from Reviewer 1, especially the request to describe the different translation-dependent quality control pathways, and explain the function and interactions between Pelo and Hbs1l, to make the manuscript more easily readable for a reader who is not an expert on the field. and make any necessary changes.

2) Please consider the suggested organizational changes from Reviewer 2 and make revisions you deem necessary.

Reviewer #1:

The study by Terrey et al. addresses a fundamental question, the consequences of defects in translation and translation-dependent quality control pathways, and shows that the NGD/NSD factors Pelo and Hbs1l are critical for cerebellar neurogenesis but expendable for survival of these neurons after development. This is a strong paper using mouse genetics and molecular biology to demonstrate an important, novel control mechanism for CNS neurogenesis and circuit function, using the cerebellum as a model. The study is important for a number of neurological disorders.

1. In the introduction, the authors should describe the different translation-dependent quality control pathways, and explain the function and interactions between Pelo and Hbs1l, to make the manuscript more easily readable for a reader who is not an expert on the field.

2. Please explain upstream regulator analysis briefly.

3. Some of the figures (e.g., Figure 2 - supplement 1: D, L and N) would benefit from an adjustment of y-axis limits to evaluate the differences between groups.

4. While this may be beyond the scope of this study (and possibly technically challenging), it would be interesting to perform ribosome profiling specifically in purified granule cells in En1-Cre, PelocKO mice at P0 or Math1-Cre; PelocKO mice around P6-P7 to further dissect the mechanisms by which translation-dependent quality control pathways regulate cell cycle exit of neuronal progenitors during neurogenesis.

Reviewer #2:

The authors first demonstrate that the ribosome rescue factors Pelo and Hbs1l are critical for embryonic and brain development. They then study mouse embryonic fibroblasts (MEFs) with deficiency of Pelo or Hbs1l and show that Loss either gene induces defects in translation elongation. Similar effects were observed when they conditionally deleted the core nonsense-mediated decay component, Upf2. Impairment of these translation-dependent quality control pathways all lead to mTORC1 activation, possibly as a compensatory response.

The data in MEFs show that Pelo and Upf2 cause a greater degree of ribosomal pausing compared to Hbs1l, but interestingly, when looking at translation efficiency, there appears to be compensatory changes in ribosomal binding and pausing does not seem to be the main driver of TE. There is data throughout the manuscript looking at genes whose changes in TE are primarily driven by the transcript level changes, in comparison to genes whose changes in TE are primarily driven by changes in ribosomal binding. It would be particularly interesting to interpret the authors 'findings regarding upstream regulators in light of these data. For example, transcription factors and regulators would be more relevant in the first group, whereas other signaling and cellular pathways might be interesting in the second group.

The degree of similarity between changes in ribosomal binding in Upf2 and Pelo mutants is striking, and future work might address the similarity between Upf2 and Hbsl1, and whether Upf2 expression is affected in Pelo and Hbs1l mutants, because of the finding that Pelo and Hbs1l interact at the post-transcriptional level.

The finding that deletion of each gene leads to mTORC1 activation is interesting. In future work, it will be interesting to look at the transcript and ribosomal binding levels of mTOR pathway genes to understand whether this effect is primarily driven by dysregulation at the transcriptional/translational level or whether it is due to alterations in cellular pathways such as stress response. The mTORC1 finding deserves to be emphasized in the manuscript because it represents the best hypothesis for a mechanism underlying the observed effects, as other potential explanations, such as ribosomal pausing, NMD, etc., seem to be less consistent with the observations.

The finding that all three genes impair cerebellar development in a roughly similar manner through alteration in progenitor proliferation and differentiation is interesting. It is also striking that mTORC1 activity as measured by phosphorylated S6 is increased in the Pelo and Upf2 conditional knockouts. This represents a potential part of mechanism underlying the cerebellar defects, and this would deserve to be featured in the main figures. In future work, the same pS6 analysis could be performed for the Hbs1l condition knockouts, to determine whether the three genes are similar in this regard as well.

I found the organization of the manuscript to be somewhat confusing and worry that it obscures some of the main points of the study. I wonder if the authors would consider starting with the inducible deletion of Hbs1l, Pelo, and Upf2 in MEFs.

It was unclear to me why the authors chose to perform ribosomal profiling from the cerebellum in the Atoh1-Cre conditional line as there was no overt phenotype in this line.

There is also a great deal of other data that is present in this paper. For example, the Hbs1l gene trap, the embryonic lethality of Hbs1l deletion, and location of ribosomal footprints within the gene. However, several of the aspects, while interesting, do not appear to serve to support the main points of the paper. Therefore, the authors could consider de-emphasizing these aspects and limiting the space in both the text and figures devoted to these findings, in order to keep the focus on their main findings.

---

## [Author Response]

Reviewer #1:The study by Terrey et al. addresses a fundamental question, the consequences of defects in translation and translation-dependent quality control pathways, and shows that the NGD/NSD factors Pelo and Hbs1l are critical for cerebellar neurogenesis but expendable for survival of these neurons after development. This is a strong paper using mouse genetics and molecular biology to demonstrate an important, novel control mechanism for CNS neurogenesis and circuit function, using the cerebellum as a model. The study is important for a number of neurological disorders.1. In the introduction, the authors should describe the different translation-dependent quality control pathways, and explain the function and interactions between Pelo and Hbs1l, to make the manuscript more easily readable for a reader who is not an expert on the field.

We have included additional information about these translation-dependent quality control pathways in the introduction.

2. Please explain upstream regulator analysis briefly.

The upstream regulator analysis tries to predict/identify potential upstream transcriptional regulators that may explain the observed expression changes. We have now included an explanation of this analysis in the manuscript.

3. Some of the figures (e.g., Figure 2 - supplement 1: D, L and N) would benefit from an adjustment of y-axis limits to evaluate the differences between groups.

We have adjusted the y-axis limits on these graphs.

4. While this may be beyond the scope of this study (and possibly technically challenging), it would be interesting to perform ribosome profiling specifically in purified granule cells in En1-Cre, PelocKO mice at P0 or Math1-Cre; PelocKO mice around P6-P7 to further dissect the mechanisms by which translation-dependent quality control pathways regulate cell cycle exit of neuronal progenitors during neurogenesis.

We agree with the reviewer that interrogating the changes in elongation/translation as cell lineages become committed would be interesting. We initially considered analyzing granule cell precursors in Atoh (Math1)-*Pelo*^cKO^ mice as these would ultimately have defects in differentiation, but we realized the interpretation of this data would be problematic since differentiation defects were detectable for anterior granule cell precursors, but not for the posterior precursors, and separating these populations is technically challenging. The isolation of granule cell precursors from En1-*Pelo*^cKO^ is even more challenging given the virtual absence of these cells at birth.

Reviewer #2:[…] I found the organization of the manuscript to be somewhat confusing and worry that it obscures some of the main points of the study. I wonder if the authors would consider starting with the inducible deletion of Hbs1l, Pelo, and Upf2 in MEFs.

We have seriously considered the reviewer’s suggestion to invert the layout of the Results section of our manuscript. However, after reflection, we still believe that the current structure is the best way to present our experiments. We think that beginning the Results section with the in vivo phenotype of *Hbs1* knockout mice illustrates the importance of *Hbs1l* for embryonic and nervous system development and thus justifies and informs the rest of the experiments in the paper. Furthermore, it is our in vivo ribosome profiling data that allows us to establish that mouse embryonic fibroblasts are a valid in vitro system for studying the function of these quality control pathways, as we observed similar translation defects in vivo and in these cells. The congruence of the *Hbs1l* mutant in vivo and MEF data led us to hypothesize that the similarities in the translation data from *Hbs1l*/*Pelo* and *Upf2* mutant MEFs would be predictive of similar in vivo CNS phenotypes when these distinct pathways were disrupted.

It was unclear to me why the authors chose to perform ribosomal profiling from the cerebellum in the Atoh1-Cre conditional line as there was no overt phenotype in this line.

Our observation that deletion of *Hbs1l* in granule cells did not evoke any cellular defects suggested that either translation defects could be less pronounced in granule cells relative to mutant embryos (which fail to develop) or that mutant granule cells could have similar levels of translation defects as mutant embryos, but are less susceptible to these defects. We performed ribosome footprint profiling on Atoh1-Cre, *Hbs1l*^cKO^ mice to distinguish between the two possibilities.

There is also a great deal of other data that is present in this paper. For example, the Hbs1l gene trap, the embryonic lethality of Hbs1l deletion, and location of ribosomal footprints within the gene. However, several of the aspects, while interesting, do not appear to serve to support the main points of the paper. Therefore, the authors could consider de-emphasizing these aspects and limiting the space in both the text and figures devoted to these findings, in order to keep the focus on their main findings.

We hope that by providing these data we may help future experimenters and studies, even if these data are not necessarily essential for the main point of our studies. For example, the *Hbs1l* gene trap allele (*Hbs1l*^tm1a^) has been used in previous studies (Liakath-Ali et al., 2018; O’Connell et al., 2019) and will likely be used in future studies, as it is commercially available. Thus, it is important to point out that even though this allele is intended to function as a null allele, there is still residual *Hbs1l* expression. This clarification is critical to avoid perpetuating potentially false negative data derived from these mice (e.g. the lack of embryonic lethality). In addition, we show that this allele can easily be converted into a complete *knockout* allele allowing experimenters to utilize this allele with its original intention. Our analysis of the *Hbs1l* gene trap allele versus the *Hbs1l* knockout also highlights the incredible sensitivity of mammalian cells to slight differences in the levels of *Hbs1l* transcripts, which cannot be extrapolated from in vitro studies. This information is likely also relevant to analysis of human mutations in various components of translational quality control pathways including *Hbs1l* (O’Connell et al., 2019).